# Panproteome-wide analysis of antibody responses to whole cell pneumococcal vaccination

Joseph J Campo[1]*, Timothy Q Le[1], Jozelyn V Pablo[1], Christopher Hung[1], Andy A Teng[1], Hervé Tettelin[2], Andrea Tate[3], William P Hanage[4], Mark R Alderson[3], Xiaowu Liang[1], Richard Malley[5], Marc Lipsitch[4,6], Nicholas J Croucher[7]*

[1]Antigen Discovery Inc, California, United States; [2]Institute for Genome Sciences, School of Medicine, University of Maryland, Baltimore, United States; [3]PATH, Seattle, United States; [4]Center for Communicable Disease Dynamics, Department of Epidemiology, Harvard TH Chan School of Public Health, Boston, United States; [5]Division of Infectious Diseases, Department of Medicine, Boston Children's Hospital and Harvard Medical School, Boston, United States; [6]Department of Immunology and Infectious Diseases, Harvard TH Chan School of Public Health, Boston, United States; [7]MRC Centre for Global Infectious Disease Analysis, Department of Infectious Disease Epidemiology, Imperial College London, London, United Kingdom

*For correspondence:
jcampo@antigendiscovery.com
(JJC);
n.croucher@imperial.ac.uk (NJC)

Competing interest: See
page 25

Reviewing editor: Bavesh D
Kana, University of the
Witwatersrand, South Africa

**Abstract** Pneumococcal whole cell vaccines (WCVs) could cost-effectively protect against a greater strain diversity than current capsule-based vaccines. Immunoglobulin G (IgG) responses to a WCV were characterised by applying longitudinally-sampled sera, available from 35 adult placebo-controlled phase I trial participants, to a panproteome microarray. Despite individuals maintaining distinctive antibody 'fingerprints', responses were consistent across vaccinated cohorts. Seventy-two functionally distinct proteins were associated with WCV-induced increases in IgG binding. These shared characteristics with naturally immunogenic proteins, being enriched for transporters and cell wall metabolism enzymes, likely unusually exposed on the unencapsulated WCV's surface. Vaccine-induced responses were specific to variants of the diverse PclA, PspC and ZmpB proteins, whereas PspA- and ZmpA-induced antibodies recognised a broader set of alleles. Temporal variation in IgG levels suggested a mixture of anamnestic and novel responses. These reproducible increases in IgG binding to a limited, but functionally diverse, set of conserved proteins indicate WCV could provide species-wide immunity.

Clinical trial registration: The trial was registered with ClinicalTrials.gov with Identifier NCT01537185; the results are available from https://clinicaltrials.gov/ct2/show/results/NCT01537185.

DOI: https://doi.org/10.7554/eLife.37015.001

## Introduction

*Streptococcus pneumoniae* (the pneumococcus), commonly carried in the nasopharynx, is an important respiratory pathogen capable of causing pneumonia, bacteraemia and meningitis. The earliest recorded pneumococcal vaccinations consisted of two doses of heat-killed pneumococci cultured from the sputum of pneumonia patients, which resulted in limited protection against pneumococcal infections for a few months after inoculation (*Maynard, 1915*). Later pneumococcal vaccines used

**eLife digest** *Streptococcus pneumoniae* is a bug that causes pneumonia and meningitis, killing around a million people each year. Vaccines now exist to protect young children against these diseases, but they are expensive and do not work against all the strains of the bacteria. This is because these shots train the body's immune system to recognize and attack the bacterium's capsule, a layer of sugars that surrounds the microbe and is often different between strains.

One possible solution could be a cheap, whole cell vaccine. These injections expose the body to genetically modified *S. pneumoniae* that do not carry the capsule. Such treatment has now been tested in a small number of people during a clinical trial.

Here, Campo et al. use a technique known as panproteome array to scan samples collected during this trial, and identify which elements the body learns to recognize when it is exposed to the genetically manipulated strain of *S. pneumoniae*. The results show that when volunteers receive this vaccine, their body targets proteins that the capsule normally shields from the immune system. Many of these proteins are very similar across all strains of *S. pneumoniae*, which means that the whole cell vaccine could potentially better protect against a broad spectrum of bacteria. However, further studies are needed to assess whether this is the case, especially in infants.

DOI: https://doi.org/10.7554/eLife.37015.002

purified capsule polysaccharides, of which there are almost 100 immunologically distinguishable variants (*Bentley et al., 2006*), termed 'serotypes'. These formulations expanded from a bivalent formulation in the 1930s (*Ekwurzel et al., 1938*) to include 23 capsular polysaccharides by the 1980s (*Mufson et al., 1985*). Such formulations afford little protection to infants, however, as polysaccharides are T-cell-independent antigens that are not efficiently recognised by the immature adaptive immune system (*Stein, 1992*). Therefore, the most commonly used pneumococcal vaccines at present are protein-polysaccharide conjugate vaccines (PCVs), currently containing up to 13 different polysaccharides, each attached to a carrier protein (*Nunes and Madhi, 2011*). These vaccines elicit protective immune responses, even in young children, which prevent nasopharyngeal carriage as well as disease (*Lee et al., 2014*).

The intrinsic disadvantage of PCVs is the limited number of serotypes against which they provide protection, resulting in serotype replacement disease that reduces their impact (*Weinberger et al., 2011*). The further expansion of their valency is limited by the complexity of their manufacture, which also makes them costly (*Miller et al., 2011*; *Ray, 2002*). Therefore, efforts have continued to develop alternative vaccines that are cheaper, generate T-cell-dependent responses to non-capsular antigens, and afford protection against all pneumococci. Whole cell-based vaccines (WCVs) present a possible solution, as they can be relatively inexpensively manufactured and present an almost full complement of antigens to the recipient's immune system. Rather than the historical precedent of killed clinical isolates, the WCV used in this study is a specifically engineered version of the unencapsulated strain *S. pneumoniae* RM200 (*Lu et al., 2010b*).

A randomised double-blind phase I safety and immunogenicity trial of this WCV in 42 healthy U.S. adults was designed to compare four cohorts (*Figure 1*). One received placebo saline injections, while the other three received 100 µg, 300 µg or 600 µg doses of the pneumococcal WCV adsorbed to aluminium hydroxide adjuvant. Each individual was given three injections 28 days apart, and serum samples were taken pre-vaccination and at 28, 56 and 84 days subsequent to the first injection to assay for immunoglobulin G (IgG) responses. To quantify the potential multiplicity of immune responses, a set of available samples were analysed with a panproteome microarray including over 2100 probes (*Croucher et al., 2017*), most of which corresponded to full-length proteins, with others representing fragments of larger polypeptides. This provides information on responses to proteins encoded by the core and accessory genome, as well as multiple variants of diverse core loci (DCL), corresponding to genes that can be identified in almost all isolates based on their location in the chromosome and the domain structure of the translated protein, but which exhibit little detectable sequence similarity across the species. In *S. pneumoniae*, four such loci encode pneumococcal surface proteins A (PspA) and C (PspC), and the zinc metalloproteases A (ZmpA) and B (ZmpB). This study therefore aimed to identify the range and types of antigens to which IgG responses were

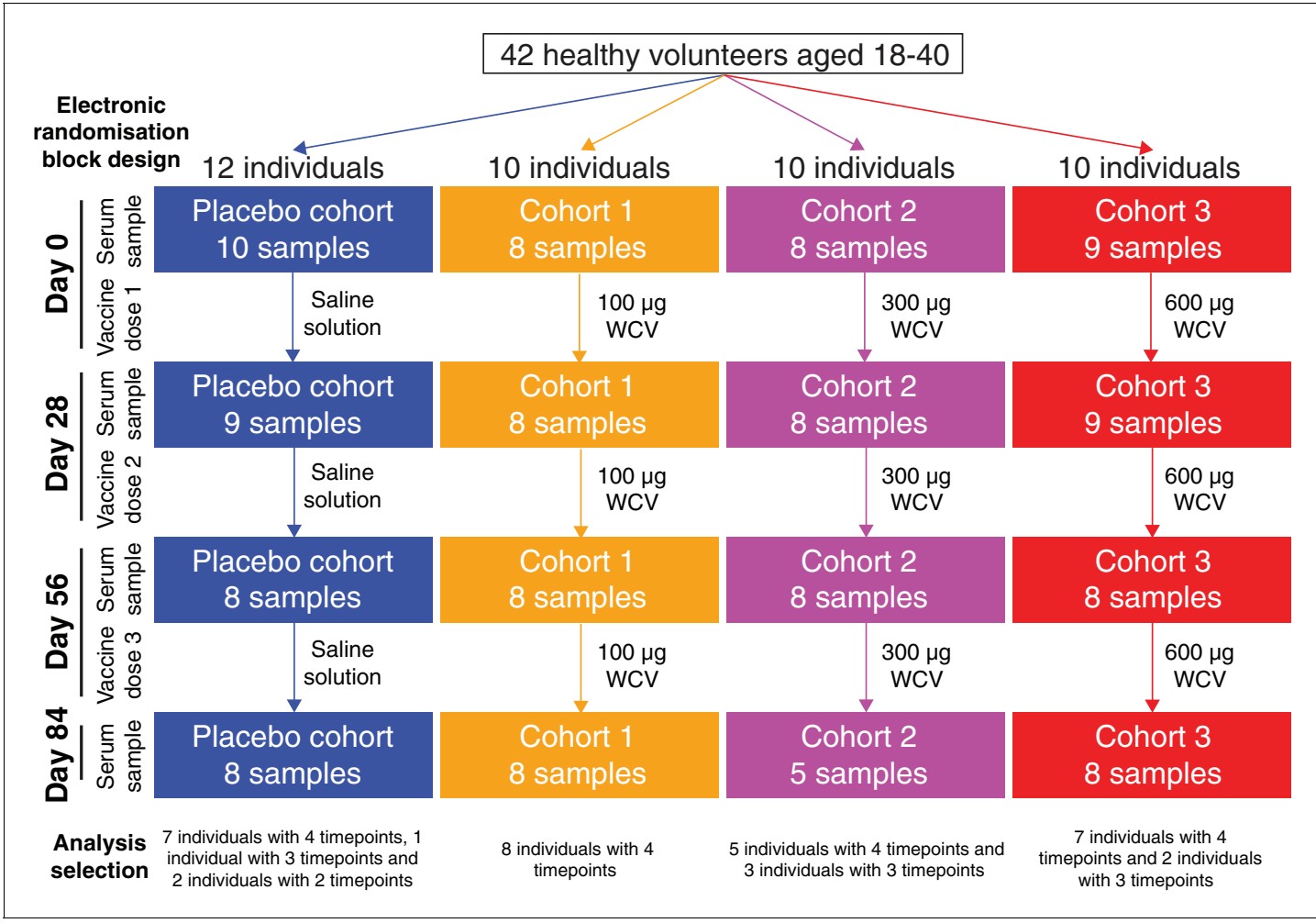

**Figure 1.** Flow chart describing the analysed samples. Forty-two healthy adult volunteers were randomly assigned to either a placebo control group, or one of three cohorts receiving different doses of the WCV, in a double-blind phase I trial. Serum samples were planned to be taken prior to the first dose, and 28 days after each of the three doses. Not all participants completed the trial, and therefore the set of available samples was limited. Those analysed in this study came from 35 of the trial participants, and are broken down by cohort and timepoint in this chart.

DOI: https://doi.org/10.7554/eLife.37015.003

The following source data and figure supplement are available for figure 1:

**Source data 1.** Sample collection data.
DOI: https://doi.org/10.7554/eLife.37015.005
**Figure supplement 1.** Construction of strain *S. pneumoniae* RM200.
DOI: https://doi.org/10.7554/eLife.37015.004

mounted following WCV administration, and how these might vary between individuals. The availability of longitudinal samples from multiple trial participants also offered the opportunity to study differences between individuals' antibody repertoires. Finally, the trial structure allowed the kinetics of responses to successive doses of different concentrations to be to quantified. These analyses should provide information on whether the WCV is likely to consistently induce antibodies capable of recognising antigens conserved across *S. pneumoniae* isolates.

## Results

### Genomic analysis of the WCV formulation

The WCV formulation contains chemically-killed *S. pneumoniae* RM200 cells (*Malley et al., 2001*), the genome sequence of which was aligned against the original progenitor, *S. pneumoniae* D39 (*Ravin, 1959*). Twenty recombinations distinguishing the pair were identified, one of which represented reversible inversion at a phase variable locus (*Croucher et al., 2014*). The generation of *S. pneumoniae* SIII-N, an intermediate genotype expressing the mucoid serotype three capsule (*Ravin, 1959*), likely accounts for eighteen of these recombinations, which span a total of 101 kb in the RM200 genome and include a recombination importing the serotype three capsule polysaccharide synthesis (*cps*) locus (*Figure 1—figure supplement 1*). RM200 was derived from Rx1, a spontaneous mutant of SIII-N that no longer expressed a capsule. Correspondingly, the *cps* locus contains two candidate mutations potentially responsible for this phenotype. An $Arg_{320}Cys$ substitution in the 6-phosphogluconate dehydrogenase protein replaces a catalytically-important arginine, which binds a pyrophosphate moiety, with a cysteine, which could interfere with the active site thiol group (*Campbell et al., 2000*). Similarly, the phosphoglucomutase protein has an $Asn_{146}Thr$ substitution that disrupts a Ser-His-Asn motif involved in divalent cation coordination, which is conserved across many orthologues (NCBI conserved domain cd05799).

A subsequently-introduced recombination represents further engineering to replace the *lytA* gene with the Janus cassette (*Sung et al., 2001*), to reduce virulence and improve the yield of cells from culture (*Berry et al., 1989*; *Lu et al., 2010b*) (*Figure 1—figure supplement 1*). Alteration of the pneumolysin toxin gene was not associated with an inferred recombination, as only three bases were substituted, in order to remove the protein's cytolytic and complement-activating activity (*Lu et al., 2010b*). This change also resulted in insertion of the pDP28 shuttle vector (accession code KJ395591) at the adjacent site (*Figure 1—figure supplement 1*).

Other than these known alterations, there were no large structural changes to the RM200 genome. The cryptic pDP1 plasmid of *S. pneumoniae* D39 was retained (*Oggioni et al., 1999*), and the large adhesin PclA was still present, but the glycoprotein PsrP, both known pneumococcal pili, and degradative zinc metalloprotease ZmpC were absent (*Croucher et al., 2017*). None of the DCL were affected by the recombinations, and consequently they were all similar to the alleles in D39. However, there was a nonsense mutation in the *pspC* gene which removed six of the eight choline-binding domains (CBDs), which may reduce the proportion of this protein attached to the cell surface.

### Stable antibody 'fingerprints' of vaccinated individuals

Overall, 130 samples were studied from 35 of the 42 trial participants: all four timepoints were analysed for 27 individuals, at least the initial and final samples were available for a further two people, and at least one timepoint was analysed for a further six people (*Figure 1*). Of these, 35 samples were from the placebo group, 32 from cohort 1 (100 µg dose), 29 from cohort 2 (300 µg dose), and 34 from cohort 3 (600 µg dose). Each sample was treated as a biological replicate. No technical replicates were included, as analysis of pilot data obtained using a smaller version of the array found reproducible differences between individuals (*Figure 2—figure supplement 1*). This suggested results were consistent between array assays of samples from the same individual, and that the study should span the maximal number of trial participants, as between-individual variation might be an important factor in understanding the response to the WCV. To visualise the relative importance of these differences between trial participants, the IgG binding across all probes for each sample was projected in two dimensions using t-distributed stochastic neighbor embedding (t-SNE; *Figure 2*), which clusters together similar sets of multidimensional data. This revealed individuals had a distinctive antibody 'fingerprint' that was generally preserved over the course of the trial, independent of vaccination dose. A similar pattern was observed when the DCL were excluded from the analysis (*Figure 2—figure supplement 2*), or when only probes from core proteins conserved across related streptococcal species were considered (*Figure 2—figure supplement 3*). Therefore, these fingerprints do not appear to reflect individuals' distinctive histories of exposure to variable proteins, but instead unique patterns of IgG responses to common antigens. These were sufficiently robust not to be disrupted by WCV administration.

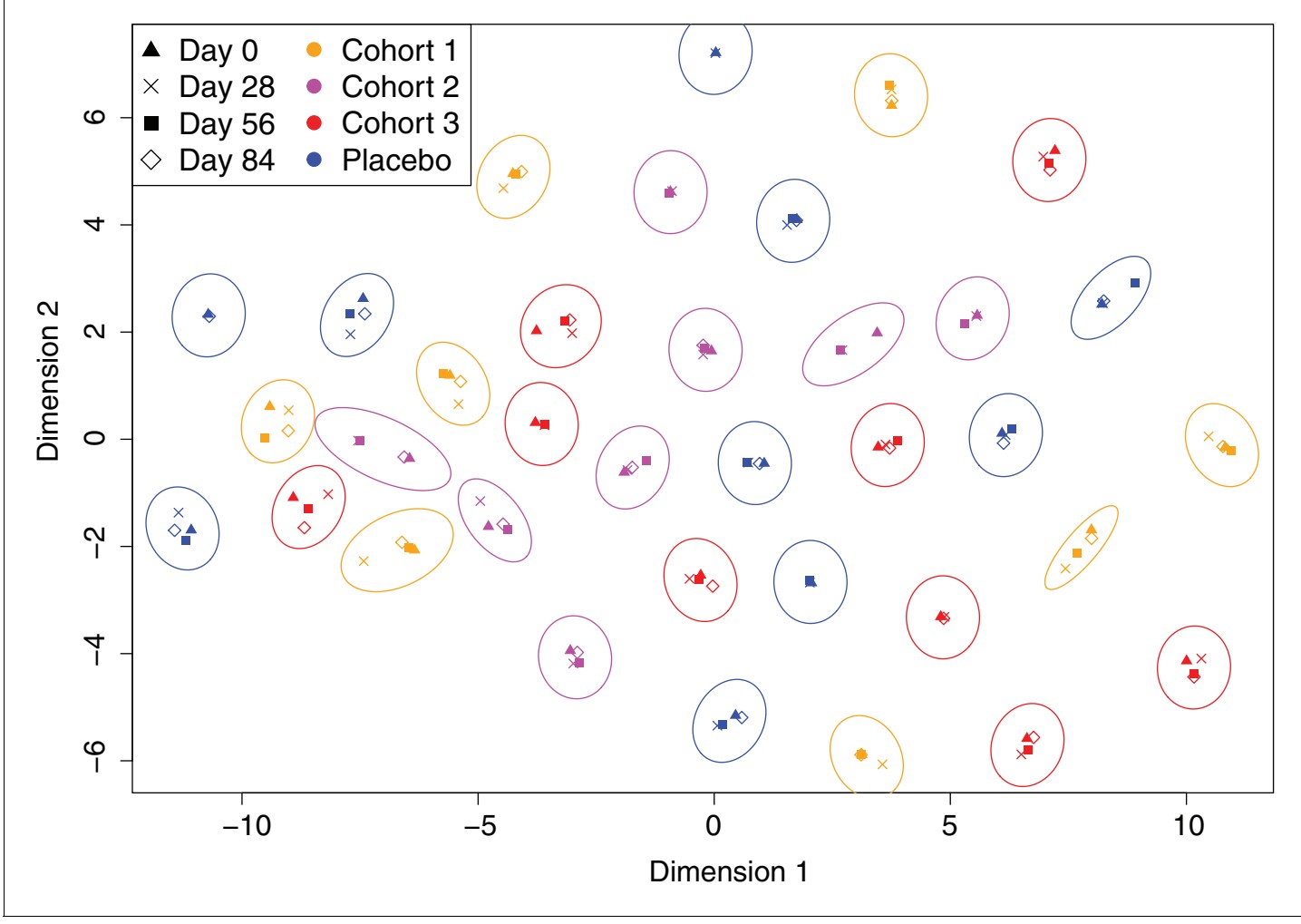

**Figure 2.** Variation in the immune profiles of trial participants. The IgG-binding data for each sample were projected across two dimensions using t-distributed stochastic neighbour embedding. Points are coloured according to the cohort of the individual contributing the sample. The shape represents the day of the trial on which the sample was collected. Ellipses surround sets of samples taken from the same individual. The separation of each ellipse shows the distinctive immune 'fingerprint' of each individual, which is maintained over the course of the trial.

DOI: https://doi.org/10.7554/eLife.37015.006

The following source data and figure supplements are available for figure 2:

**Source data 1.** Panproteome array IgG binding data.

DOI: https://doi.org/10.7554/eLife.37015.010

**Figure supplement 1.** Reproducibility of probe signals.

DOI: https://doi.org/10.7554/eLife.37015.007

**Figure supplement 2.** Evidence of distinct antibody fingerprints of trial participants from non-DCL probes.

DOI: https://doi.org/10.7554/eLife.37015.008

**Figure supplement 3.** Evidence of distinct antibody fingerprints of trial participants from proteins conserved across *S. pneumoniae*, *S. pseudopneumoniae* and *S. mitis.*

DOI: https://doi.org/10.7554/eLife.37015.009

Despite this starting variation between trial participants, there was no overall significant difference in individuals' median IgG binding to *S. pneumoniae* proteins between cohorts before vaccination (*Figure 4—figure supplement 1A*; Kruskal-Wallis test, $N = 29$, $\chi^2 = 0.20$, df = 3, p = 0.98). Changes in IgG binding between the start of the trial, day 0, and the end, day 84, were quantified as $\Delta_{0 \to 84}$ (*Figure 4—figure supplement 1B*). The cohorts did not differ in the distribution of median $\Delta_{0 \to 84}$ per individual (Kruskal-Wallis test, $N = 29$, $\chi^2 = 3.17$, df = 3, p = 0.37), indicating that the WCV did not raise IgG binding to a high proportion of *S. pneumoniae* proteins across all members

of the vaccinated cohorts. Restricting this analysis to the antibody-binding targets (ABTs), defined as those proteins associated with high IgG binding in the pre-vaccination sample (*Croucher et al., 2017*), also failed to detect overall heterogeneity in the distribution of individuals' median $\Delta_{0\rightarrow84}$ between the groups (Kruskal-Wallis test, $N = 29$, $\chi^2 = 4.63$, df = 3, p = 0.20; *Figure 4—figure supplement 1C*).

However, an ANOVA test of the fits of linear mixed effects models of $\Delta_{0\rightarrow84}$ across probes (considered fixed effects) and individuals (considered random effects) found a significant improvement when cohort was also included as a fixed effect ($\chi^2 = 8.96$, df = 3, p = 0.030; see Materials and methods). To test whether this reflected biologically uninformative changes in the responses to poorly immunogenic proteins, this analysis was repeated using only the 1,584 'immunoreactive' probes, to which an IgG binding of at least one (i.e. twice the background level) was detected across the dataset. This found a more significant effect of including cohort as a fixed effect ($\chi^2 = 11.8$, df = 3, p = 0.0081), indicating there was a statistically, and potentially biologically, significant difference in the distribution of $\Delta_{0\rightarrow84}$ between cohorts. Hence the WCV did have a detectable effect on vaccinated cohorts when considering per-probe, rather than per-individual, data. Combined with the observation that unique antibody fingerprints were maintained throughout the trial, this suggests the WCV either elevated responses to only a subset of proteins, or uniformly boosted multiple anamnestic responses in only a subset of individuals.

## WCV elicits elevated IgG to a specific minority of proteins

Comparing the within-cohort medians of pre-vaccine IgG binding and $\Delta_{0\rightarrow84}$ across all probes provided a simple approach to judging the relative contributions of these alternative explanations. If there were a broad response in only some individuals, the median $\Delta_{0\rightarrow84}$ values would not be expected to vary much across probes, except for a general rise with higher pre-vaccine antibody binding, if this correlated with the strength of immunological memory. Alternatively, a strong response to only a subset of probes across individuals should result in much greater variation in within-cohort median $\Delta_{0\rightarrow84}$. The plots revealed a statistically significant positive correlation between pre-vaccine-binding strength and post-vaccine rise for the three vaccinated groups (Pearson correlation coefficient, p < $10^{-16}$ for each), as expected for a uniform rise in IgG binding if many anamnestic responses were upregulated. By contrast, a negative correlation was evident in the placebo recipients, possibly representing regression to the mean (*Figure 3A–D*). Yet these general increasing trends for the vaccinated cohorts were small compared to the large increases in IgG binding to a particular subset of antigens. This indicates the post-vaccine changes primarily represented strong responses to a subset of proteins.

## ABTs are enriched among immunogenic WCV proteins

These plots indicated many probes associated with larger increases in IgG binding were already strongly recognised by adaptive immunity pre-WCV. This suggested WCV antigens, which were defined as being associated with $\Delta_{0\rightarrow84}$ above the empirically-derived threshold $\Delta_{0\rightarrow84}$ of 0.2 (*Figure 4—figure supplement 2*), would be enriched for ABTs, those proteins to which IgG binding was already high pre-WCV. ABT probes accounted for 23 of 47 probes rising above the threshold in cohort 1 (48.9%, a 5.95-fold enrichment; Fisher's exact test, $N = 2,343$, p = $6.11\times10^{-9}$), 17 of 43 probes in cohort 2 (39.5%, a 4.06-fold enrichment; Fisher's exact test, $N = 2,343$, p = $2.37\times10^{-5}$) and 60 of 129 probes in cohort 3 (46.5%, a 5.40-fold enrichment; Fisher's exact test, $N = 2,343$, p < $10^{-16}$). These tests consistently showed ABTs were disproportionately associated with increased IgG binding post-vaccination, although the absolute numbers indicate many ABTs did not trigger a substantial response. The distinction between those proteins to which there was little reaction, and those eliciting large increases in IgG binding post-WCV, was generally consistent both within cohorts and between vaccinated cohorts (*Figure 3E–F*), suggesting these specific reactions to ABTs were not stochastic, nor driven by variation between individuals' pre-existing antibody profiles.

These results may represent WCV-induced responses being limited to those components of the panproteome that are similar to proteins expressed by RM200. To test this hypothesis, an amino acid sequence identity threshold of at least 90% between array and RM200 proteins was empirically determined from the distribution of pairwise sequence identities (*Figure 4—figure supplement 4*). Excluding all DCL variants, 1602 proteins (corresponding to 1647 probes) could be matched to

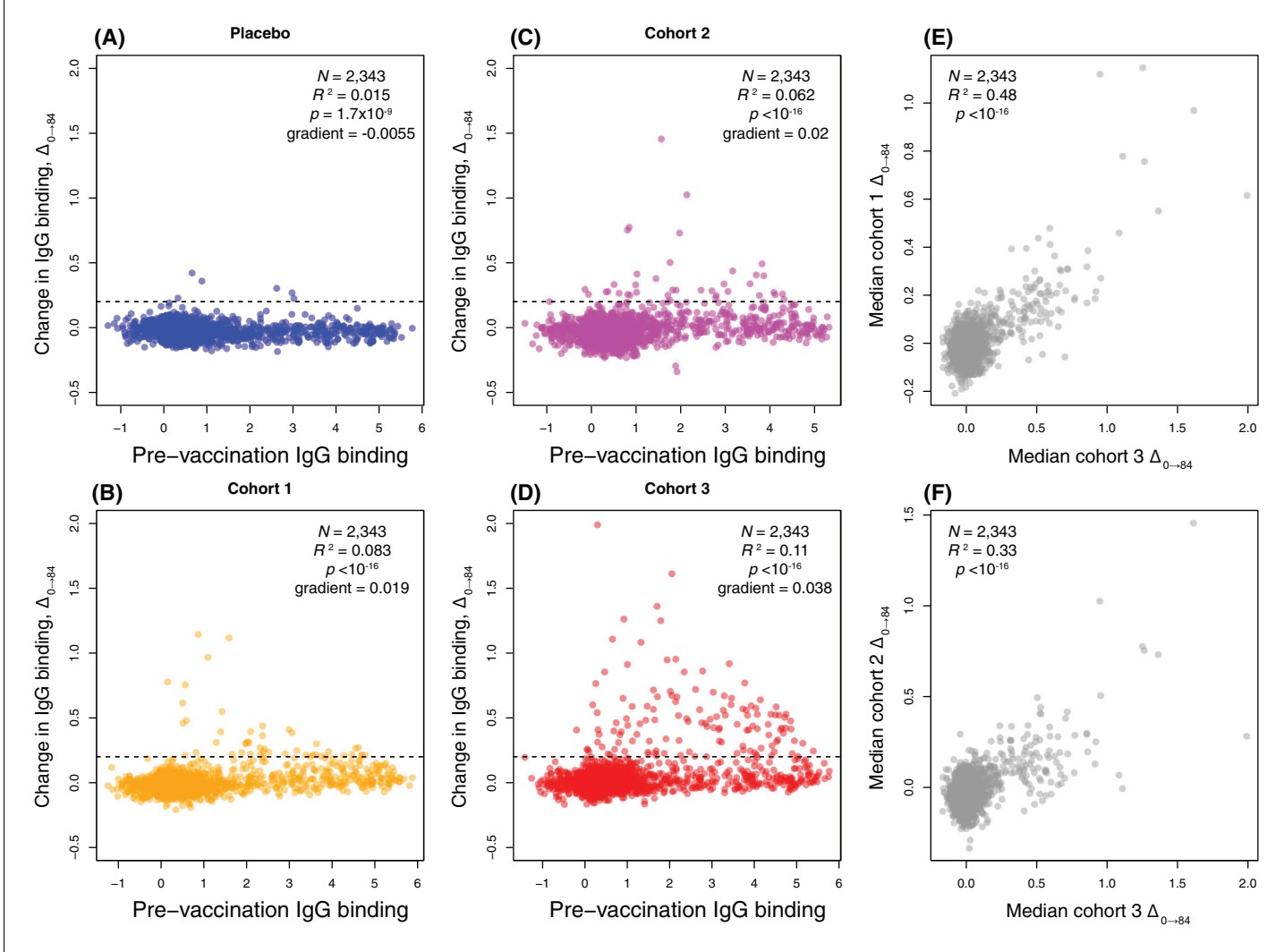

**Figure 3.** Scatterplots showing the within-cohort median change in IgG binding to each probe over the course of the trial ($\Delta_{0 \to 84}$) relative to the within-cohort median pre-vaccination IgG binding. (A–D) This relationship is shown for each probe on the proteome array using data from the 29 individuals with samples taken at the start and end of the trial. These are split into (A) the placebo group, (B) cohort 1 (100 μg doses), (C) cohort 2 (300 μg doses), and (D) cohort 3 (600 μg doses). The horizontal dashed line indicates the empirically-determined threshold for defining probes having undergone a substantial change in IgG binding during the trial (*Figure 4—figure supplement 2*). The Pearson's correlation statistics and gradient of the best-fitting linear model are annotated on each panel. (E–F) The relationship between the median $\Delta_{0 \to 84}$ for each probe is shown for (E) a comparison of cohorts 1 and 3, and (F) a comparison of cohorts 2 and 3.
DOI: https://doi.org/10.7554/eLife.37015.011

RM200 proteins with a similarity above the 90% threshold, while 455 proteins (corresponding to 476 probes) lacked a close orthologue in RM200. Using these two categories to represent epitopes present and absent from the WCV respectively, a comparison of the median $\Delta_{0 \to 84}$ using a Wilcoxon rank sum test found no significant difference in the post-WCV changes in IgG binding in cohort three between these sets of probes (*Figure 4A*, left panel; Wilcoxon rank sum test, $N = 2,123$, $W = 403523$, p = 0.33). However, an analogous test restricted to ABTs found significantly stronger antibody rises ($\Delta_{0 \to 84}$) associated with the 70 ABTs with a close orthologue in RM200 (corresponding to 87 probes) relative to the 25 ABTs absent from the strain (corresponding to 40 probes; *Figure 4A*, middle panel; Wilcoxon rank sum test, $N = 127$, $W = 970$, p = $6.5 \times 10^{-5}$). This suggests a vaccine-induced IgG response that primarily targets naturally immunogenic proteins expressed by the WCV, although some ABTs present in the WCV still did not elicit substantially increased antibody binding.

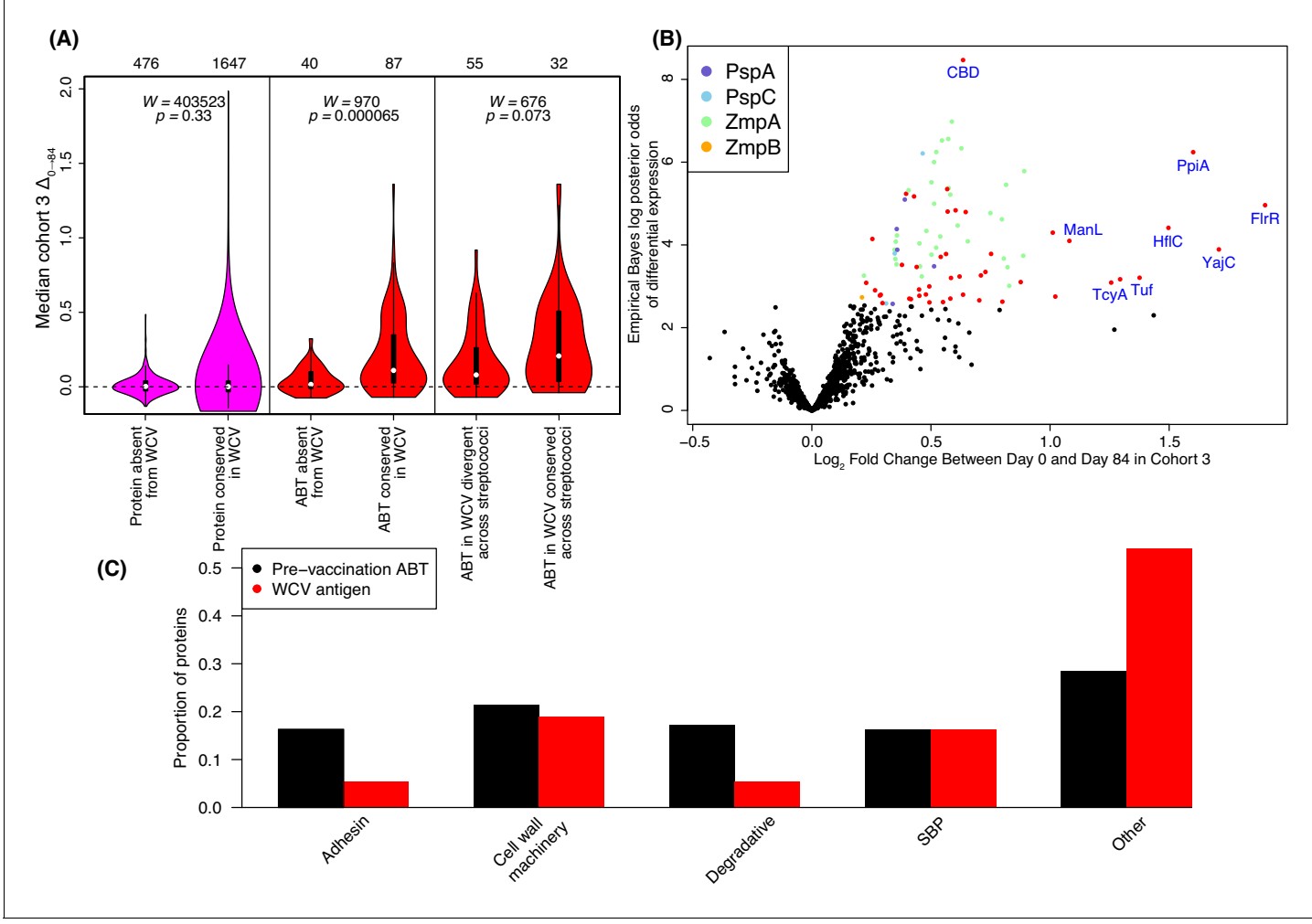

**Figure 4.** Identification of WCV antigens from changes in IgG binding. (A) Violin plots showing the median $\Delta_{0\to84}$ within cohort three for non-DCL array probes, split by whether they represent proteins with close orthologues in the WCV strain RM200 (defined as $\geq$90% amino acid identity) or not. The central panel shows the same comparison, but further constraining the dataset to probes representing the proteins classified as ABTs based on high pre-vaccination IgG binding. The right panel stratifies the 87 probes corresponding to ABTs with close orthologues in RM200 by whether the protein on the array was also conserved with $\geq$90% sequence identity in *S. mitis* and *S. pseudopneumoniae* (*Supplementary file 1*). The number of probes in each category is shown at the top of the plot. The results of Wilcoxon rank sum tests (W) and significance (p) are annotated on each panel. (B) Volcano plot showing the statistical and biological significance of changes in IgG binding following WCV administration in cohort 3. The horizontal axis shows the fold change in IgG binding between the day zero and day 84 samples from cohort three on a base two logarithmic scale. The vertical axis shows the B statistic from the empirical Bayes analysis, representing the natural logarithm of the odds ratio of differential IgG binding between cohort three and the placebo group. Points corresponding to array probes with a Benjamini-Hochberg corrected *p* value below 0.05 are coloured red, unless they represent a variant of PspA, PspC, ZmpA, or ZmpB. (C) Functional categorisation of antigens. The distribution of ABTs, defined as eliciting high IgG in the pre-vaccination samples (*Croucher et al., 2017*), and the WCV antigens identified by either the eBayes or LMM analyses (*Supplementary file 2*) are compared across different functional categories.

DOI: https://doi.org/10.7554/eLife.37015.012

The following figure supplements are available for figure 4:

**Figure supplement 1.** Variation in IgG binding between individuals.
DOI: https://doi.org/10.7554/eLife.37015.013

**Figure supplement 2.** Determining the threshold for identifying substantial changes in IgG binding between timepoints.
DOI: https://doi.org/10.7554/eLife.37015.014

**Figure supplement 3.** Pre-vaccination IgG binding in cohort three to protein categories defined in *Figure 4*.
DOI: https://doi.org/10.7554/eLife.37015.015

**Figure supplement 4.** Sequence identity between proteomes of different species.
DOI: https://doi.org/10.7554/eLife.37015.016

**Figure supplement 5.** Effect of divergence between array sequences and RM200 proteins on detected changes in IgG binding.

*Figure 4 continued on next page*

*Figure 4 continued*

DOI: https://doi.org/10.7554/eLife.37015.017

**Figure supplement 6.** Volcano plot showing the statistical and biological significance of changes in IgG binding following WCV administration in cohort 3, when the DCL probes are excluded from the eBayes analysis.

DOI: https://doi.org/10.7554/eLife.37015.018

**Figure supplement 7.** Venn diagrams comparing the results of the empirical Bayes analysis, linear mixed effects models, and empirically-derived threshold applied to the $\Delta_{0 \to 84}$ values shown on the scatterplots in *Figure 3*.

DOI: https://doi.org/10.7554/eLife.37015.019

**Figure supplement 8.** Change in IgG binding during the trial for solute- binding proteins defined as antibody-binding targets on the basis of high pre-vaccination IgG binding.

DOI: https://doi.org/10.7554/eLife.37015.020

**Figure supplement 9.** Change in IgG binding during the trial for cell wall synthesis and processing machinery proteins defined as antibody-binding targets on the basis of high pre-vaccination IgG binding.

DOI: https://doi.org/10.7554/eLife.37015.021

**Figure supplement 10.** Maximum likelihood phylogenetic analysis of the *S.*

DOI: https://doi.org/10.7554/eLife.37015.022

**Figure supplement 11.** Changes in IgG binding to penicillin-binding protein variants.

DOI: https://doi.org/10.7554/eLife.37015.023

**Figure supplement 12.** Change in IgG binding during the trial for adhesin proteins defined as antibody-binding targets on the basis of high pre-vaccination IgG binding.

DOI: https://doi.org/10.7554/eLife.37015.024

**Figure supplement 13.** Maximum likelihood phylogenetic analysis of the *S.*

DOI: https://doi.org/10.7554/eLife.37015.025

**Figure supplement 14.** Change in IgG binding to large antigenic structures.

DOI: https://doi.org/10.7554/eLife.37015.026

**Figure supplement 15.** Change in IgG binding during the trial for surface-associated degradative enzymes defined as antibody-binding targets on the basis of high pre-vaccination IgG binding.

DOI: https://doi.org/10.7554/eLife.37015.027

## Post-WCV changes are not artefacts of commensal carriage or array design

A possible confounding factor in this analysis is the higher level of sequence conservation expected for those proteins that are similar in RM200 and the array. This may mean these proteins' greater $\Delta_{0 \to 84}$ values could be the result of the array being more sensitive to IgG adapted to recognising comparatively invariant antigens. However, a comparison of $\Delta_{0 \to 84}$ and sequence divergence between the WCV and proteins on the array failed to find evidence of a general correlation between $\Delta_{0 \to 84}$ and lower sequence divergence between the array proteins and RM200 (*Figure 4—figure supplement 5*). Immune responses to more conserved proteins could also result from encounters with the related commensal streptococci *Streptococcus mitis* and *Streptococcus pseudopneumoniae*. However, the overall distribution of $\Delta_{0 \to 84}$ for ABTs conserved in these other species was not significantly higher than to those ABTs exhibiting greater interspecies divergence (*Figure 4A*, right panel). There was also no significant difference in the pre-vaccination IgG binding to ABTs in the WCV (Wilcoxon rank sum test, $W = 1455$, $p = 0.20$), nor to those conserved in *S. mitis* and *S. pseudopneumoniae* (Wilcoxon rank sum test, $W = 937$, p = 0.62; *Figure 4—figure supplement 3*), suggesting recent immune priming through asymptomatic carriage of naturally unencapsulated streptococci is unlikely to play a major role in determining the immunity induced by WCV.

## Consistent identification of antigens by complementary statistical analyses

Rather than relying on a threshold value of $\Delta_{0 \to 84}$, two approaches commonly used in the analysis of transcriptome data were employed to identify which proteins elicited statistically significant IgG responses. Only the 1584 immunoreactive probes were included in each, to avoid analysis of changes unlikely to have biological relevance. Firstly, an empirical Bayes (eBayes) analysis, which calculates probabilities relative to a prior distribution estimated from the dataset itself, was applied to pairwise contrasts of $\Delta_{0 \to 84}$ across immunoreactive proteins in each of the vaccinated cohorts against

the same metric in the placebo group. By only using two timepoints, data from 29 individuals could be included. A Benjamini-Hochberg correction adjusting for 1584 tests, with an expected false discovery rate of 0.05 (*Supplementary file 2*), identified 88 probes as having significant $\Delta_{0\to84}$ values in the comparison of the placebo group with cohort 3, who received the highest dose of the WCV. The magnitudes of $\Delta_{0\to84}$ in this cohort are compared with the statistical significance of this change relative to the placebo group in *Figure 4B*. After the correction for multiple testing, no probes were associated with significant changes in IgG binding in the comparisons of the placebo group with cohorts 1 or 2.

Secondly, a linear mixed effects model (LMM) was used to interpret the IgG binding at each timepoint as a noisy linear response to random effects, corresponding to the different trial participants, and two sets of fixed effects, corresponding to the dose of vaccine and number of injections received. This maximised the information extracted from the complete longitudinal time series, but was limited to the 20 individual trial participants in the three vaccinated cohorts for whom these data were available. Likelihood ratio tests were then used to identify proteins that elicited IgG responses that significantly increased with dose or with number of immunisations. After a Benjamini-Hochberg correction for multiple testing, none of the proteins showed a significant dose-response. This is likely due to the anomalous behaviour of cohort 2, in which the overall changes in IgG were smaller than those for cohort 1 (*Figure 3*). By contrast, 127 probes showed a significant change in IgG binding with repeated immunisations, with all but one of these probes exhibiting increased IgG binding over the course of the trial (*Supplementary file 2*). Of these, 77 were also significant in the eBayes analysis of $\Delta_{0\to84}$ values (*Figure 4—figure supplement 7*).

Therefore, this combination of using all individuals for whom initial and final paired samples were available, as well as the full longitudinal data for vaccinated individuals, identified a total of 138 probes to which IgG binding changed significantly (*Table 1*). These were highly consistent with the 129 probes identified by applying the empirically-derived threshold to $\Delta_{0\to84}$ values from cohort three in *Figure 3D* (*Figure 4—figure supplement 7*). Sixty of the 138 probes represented DCL variants; to test whether the inclusion of this allelic variation affected the model fitting or multiple correction testing, the eBayes and LMM analyses were repeated using the 1384 immunoreactive non-DCL probes. This found 74 of the 78 non-DCL probes were still associated with significant rises in IgG binding if the DCL were excluded from the analyses, with no extra hits being detected, indicating these results were generally robust to changes in the set of probes assayed on the array (*Supplementary file 2* and *Figure 4—figure supplement 6*). Accounting for those probes that represented different regions of large proteins, the 138 probes came from 112 distinct protein sequences, of which 59.8% were identified by both eBayes and LMM approaches (*Figure 4—figure supplement 7*). Hence the statistical analyses provide a consistent view of the most immunogenic proteins in the WCV.

Seventy-one of the 112 putative antigens were from the 208 ABTs previously defined using the pre-vaccination timepoint (63.4%; a 6.16-fold enrichment) (*Croucher et al., 2017*). This represented a significant enrichment of ABTs relative to the other immunoreactive proteins (Fisher's exact test, $N = 1,443$, $p < 10^{-16}$). When limited to the 1062 immunoreactive proteins exhibiting at least 90% similarity between the WCV and the array, ABTs accounted for 39 of 71 significant increases in IgG binding (54.9%; a 16.01-fold enrichment), and 34 of the 991 proteins not associated with such an increase (3.43%; Fisher's exact test, $N = 1,062$, $p < 10^{-16}$). Hence this highlights not just the importance of ABTs in the immune response to the WCV, but also the failure of some ABTs to trigger significantly elevated IgG binding, despite the array being appropriately constructed to detect such a response (*Supplementary file 2*). Therefore, the subset of proteins exhibiting significant and reproducible increases in binding following WCV administration were functionally characterised to identify the properties associated with immunogenic proteins.

## Diverse functional characteristics of proteins eliciting IgG responses

The most statistically significant increase in IgG binding identified by the eBayes analysis was the only probe that did not correspond to a particular protein, but instead was an oligomer of choline-binding domains (CBDs). This common motif, by which multiple pneumococcal proteins adhere to the cell surface via the cell wall polysaccharide, was found to be a protein domain common in ABTs (*Croucher et al., 2017*) (*Figure 4B*). Previous data suggested such domains could be immunogenic

**Table 1.** Description of WCV antigenic proteins, represented by probes associated with significantly increased IgG binding following WCV administration by either the empirical Bayes or linear mixed effects analyses (***Supplementary file 2***).

Each protein is listed (either by common name, or COG assignation), alongside a functional annotation and the coding sequence for the orthologue in *S. pneumoniae* D39, where it could be identified. IgG-binding values were aggregated across all probes corresponding to the named protein that were associated with a significant change. Values are summarised as a median, with the interquartile range in parentheses. The columns show the median initial IgG binding to the probes across all cohorts, and the $\Delta_{0\to84}$ values for each of the four cohorts separately. The final column identifies in which analyses the set of probes corresponding to the named protein were found to be associated with elevated IgG binding.

| Protein name | Functional annotation | *S. pneumoniae* D39 coding sequence | IgG binding at day 0 | $\Delta_{0\to84}$ in placebo group | $\Delta_{0\to84}$ in cohort 1 | $\Delta_{0\to84}$ in cohort 2 | $\Delta_{0\to84}$ in cohort 3 | Statistical analysis |
|---|---|---|---|---|---|---|---|---|
| AatB | Amino acid ABC transporter solute-binding protein AatB | SPD_1328 | 3.568 (2.920–4.247) | −0.078 (−0.126–0.105) | 0.105 (−0.004–0.204) | 0.096 (0.076–0.214) | 0.285 (0.155–0.747) | Both |
| AcoC | Dihydrolipoyllysine-residue acetyltransferase component of pyruvate dehydrogenase complex AcoC | SPD_1026 | 0.722 (0.381–1.346) | −0.149 (−0.180 - −0.053) | 0.021 (−0.049–0.475) | −0.003 (−0.062–0.019) | 0.463 (0.252–0.936) | Both |
| AliA | ABC oligopeptide transporter solute-binding protein AliA | SPD_0334 | 2.803 (2.141–3.679) | −0.085 (−0.149 - −0.008) | 0.042 (−0.076–0.086) | 0.203 (0.060–0.693) | 0.323 (0.179–0.576) | Both |
| AliB | Oligopeptide ABC transporter solute-binding protein AliB | SPD_1357 | 4.098 (3.457–4.553) | −0.018 (−0.111–0.094) | 0.142 (0.038–0.229) | 0.159 (0.015–0.305) | 0.159 (0.082–0.242) | Linear mixed effects |
| AmiA | Oligopeptide ABC transporter solute-binding protein AmiA | SPD_1671 | 3.680 (2.954–4.122) | −0.007 (−0.073–0.079) | 0.216 (0.068–0.489) | 0.343 (0.056–0.369) | 0.439 (0.259–0.532) | Linear mixed effects |
| BgaA | Beta-galactosidase BgaA | SPD_0562 | 3.583 (2.453–4.350) | −0.034 (−0.060–0.030) | 0.031 (−0.030–0.130) | 0.235 (0.069–0.328) | 0.399 (0.194–0.601) | Both |
| CBD | Choline-binding domain | - | 0.937 (0.579–1.498) | −0.106 (−0.118 - −0.027) | 0.086 (0.031–0.135) | 0.204 (0.157–0.215) | 0.589 (0.357–0.658) | Both |
| CibA | Competence-induced bacteriocin A | SPD_0133 | 0.678 (0.566–1.072) | 0.007 (−0.049–0.061) | 0.756 (0.173–1.957) | 0.755 (0.363–1.435) | 1.261 (0.228–2.579) | Linear mixed effects |
| CLS00168 | Uncharacterised membrane protein | SPD_0093 | 2.540 (1.470–3.075) | −0.069 (−0.116–0.040) | 0.238 (0.136–0.631) | 0.031 (0.005–0.039) | 0.422 (0.233–0.601) | Linear mixed effects |
| CLS00229 | Uncharacterised membrane protein | SPD_0174 | 0.204 (0.132–0.397) | 0.004 (−0.124–0.043) | 0.127 (0.021–0.291) | 0.032 (0.000–0.146) | 0.602 (0.263–1.727) | Both |
| CLS00234 | Uncharacterised lipoprotein | SPD_0179 | 3.575 (2.996–4.215) | −0.028 (−0.060–0.016) | 0.035 (0.004–0.114) | −0.033 (−0.066–0.145) | 0.522 (0.284–0.829) | Both |
| CLS00386 | Uncharacterised membrane protein | SPD_0342 | 3.870 (3.551–4.418) | −0.011 (−0.050–0.049) | 0.207 (0.148–0.301) | 0.160 (0.054–0.322) | 0.370 (0.180–0.517) | Both |
| CLS01171 | Conjugative element protein | - | 1.156 (0.968–1.327) | −0.049 (−0.139 - −0.016) | 0.020 (−0.048–0.108) | 0.042 (−0.050–0.058) | 0.146 (0.072–0.290) | Empirical Bayes |
| CLS01337 | Uncharacterised membrane protein | SPD_1380 | 1.123 (0.603–1.572) | −0.061 (−0.168–0.053) | 0.187 (0.007–0.987) | 0.132 (0.038–0.183) | 0.912 (0.708–1.116) | Both |
| CLS01383 | Uncharacterised membrane protein | SPD_1429 | 4.805 (3.976–5.309) | −0.015 (−0.058–0.006) | 0.190 (0.123–0.281) | 0.088 (0.043–0.340) | 0.226 (0.194–0.504) | Linear mixed effects |
| CLS01446 | Sialic acid and N-acetylmannosamine ABC transporter permease | SPD_1500 | 0.033 (−0.086–0.080) | −0.118 (−0.175 - −0.074) | 0.074 (−0.039–0.175) | 0.073 (0.062–0.126) | 0.187 (−0.004–0.259) | Linear mixed effects |

*Table 1 continued on next page*

*Table 1 continued*

| Protein name | Functional annotation | *S. pneumoniae* D39 coding sequence | IgG binding at day 0 | $\Delta_{0\to84}$ in placebo group | $\Delta_{0\to84}$ in cohort 1 | $\Delta_{0\to84}$ in cohort 2 | $\Delta_{0\to84}$ in cohort 3 | Statistical analysis |
|---|---|---|---|---|---|---|---|---|
| CLS01563 | ROK-family transcriptional repressor protein | - | 1.502 (1.030–1.894) | −0.098 (−0.141 - −0.039) | 0.049 (−0.045–0.499) | 0.028 (−0.003–0.044) | 0.248 (0.172–0.653) | Linear mixed effects |
| CLS01820 | Uncharacterised exported protein | SPD_1928 | 0.118 (−0.012–0.276) | −0.045 (−0.064–0.075) | 0.092 (0.012–0.152) | 0.118 (0.048–0.123) | 0.261 (0.102–0.460) | Both |
| CLS02831 | Bacteriocin ABC processing efflux pump | SPD_1752 | 1.000 (0.685–1.397) | −0.013 (−0.124–0.038) | 0.135 (0.058–0.236) | 0.110 (0.074–0.178) | 0.375 (0.299–0.485) | Both |
| CLS02897 | Uncharacterised membrane protein (fragment) | - | 0.683 (0.469–0.887) | −0.066 (−0.198–0.131) | 0.085 (−0.003–0.136) | 0.120 (0.061–0.248) | 0.175 (0.075–0.243) | Linear mixed effects |
| DnaK | Chaperone protein DnaK | SPD_0460 | 0.275 (0.178–0.657) | −0.041 (−0.219–0.013) | 0.059 (−0.015–0.100) | 0.088 (0.001–0.236) | 0.375 (0.020–2.027) | Linear mixed effects |
| Dpr | DNA protection during starvation stress resistance protein Dpr | SPD_1402 | 0.988 (0.467–1.541) | 0.038 (−0.056–0.223) | 0.114 (0.034–0.219) | 0.192 (−0.031–0.462) | 0.149 (0.066–0.366) | Linear mixed effects |
| EzrA | Septation ring formation regulator EzrA | SPD_0710 | 2.400 (1.520–3.755) | −0.016 (−0.041–0.002) | 0.195 (0.096–0.258) | 0.100 (−0.014–0.120) | 0.560 (0.140–0.646) | Both |
| FabE | Biotin carboxyl carrier protein of acetyl-CoA carboxylase FabE | SPD_0386 | −0.206 (−0.459–0.513) | −0.021 (−0.119–0.031) | 0.187 (0.033–0.274) | 0.270 (0.100–0.295) | 0.408 (0.014–0.716) | Linear mixed effects |
| FrlR | HTH-type transcriptional repressor FrlR | SPD_0064 | 0.445 (0.205–0.942) | −0.039 (−0.085–0.071) | 0.615 (0.117–1.452) | 0.283 (0.231–0.730) | 1.986 (1.454–2.437) | Both |
| FruA | Fructose PTS transporter protein FruA | SPD_0773 | 1.044 (0.568–1.389) | −0.029 (−0.125 - −0.002) | 0.480 (0.118–0.817) | 0.335 (0.322–0.360) | 0.591 (0.406–0.770) | Linear mixed effects |
| GalT | Galactose-1-phosphate uridylyltransferase GalT | SPD_1633 | 0.295 (0.211–0.494) | −0.024 (−0.055–0.047) | −0.110 (−0.220 - −0.034) | −0.068 (−0.082–0.012) | −0.018 (−0.129 - −0.005) | Linear mixed effects |
| GlnH | ABC glutamine transporter solute-binding protein GlnH | SPD_1226 | 2.912 (1.858–4.006) | 0.047 (−0.094–0.099) | 0.192 (0.059–0.274) | 0.289 (0.204–0.393) | 0.770 (0.336–1.163) | Both |
| GlnPH1 | Glutamine ABC transporter permease GlnPH1 | SPD_0412 | 1.029 (0.573–1.920) | −0.026 (−0.046–0.169) | 0.178 (0.133–0.387) | 0.278 (0.105–0.447) | 0.495 (0.258–0.576) | Both |
| GlnPH4 | Amino acid ABC transporter permease GlnPH4 | SPD_1098 | 4.006 (3.344–4.515) | 0.083 (−0.016–0.317) | 0.089 (0.019–0.155) | 0.154 (0.006–0.296) | 0.269 (0.173–0.367) | Linear mixed effects |
| GroEL | Chaperonin GroEL | SPD_1709 | 2.252 (1.129–3.083) | −0.057 (−0.101–0.010) | 0.396 (0.194–0.571) | 0.068 (−0.033–0.856) | 0.426 (0.107–0.963) | Both |
| HtrA | Surface-associated serine protease HtrA | SPD_2068 | 2.617 (2.002–3.293) | 0.011 (−0.000–0.061) | 0.262 (0.102–0.578) | 0.052 (0.029–0.297) | 0.519 (0.220–0.816) | Both |
| HylD | Efflux pump protein HylD | SPD_0686 | 4.590 (4.065–4.848) | −0.045 (−0.091–0.011) | 0.177 (−0.027–0.295) | 0.259 (0.083–0.264) | 0.319 (0.178–0.489) | Linear mixed effects |
| LemA | Uncharacterised membrane protein LemA | SPD_1139 | 0.473 (0.151–1.041) | −0.003 (−0.103–0.103) | 0.170 (0.110–0.457) | 0.300 (0.004–0.392) | 0.855 (0.635–2.035) | Both |
| LytA | Lytic amidase A | SPD_1737 | 3.547 (2.672–3.735) | 0.032 (−0.094–0.103) | 0.098 (0.029–0.221) | 0.021 (−0.012–0.066) | 0.158 (0.067–0.190) | Linear mixed effects |

*Table 1 continued on next page*

*Table 1 continued*

| Protein name | Functional annotation | *S. pneumoniae* D39 coding sequence | IgG binding at day 0 | $\Delta_{0\rightarrow84}$ in placebo group | $\Delta_{0\rightarrow84}$ in cohort 1 | $\Delta_{0\rightarrow84}$ in cohort 2 | $\Delta_{0\rightarrow84}$ in cohort 3 | Statistical analysis |
|---|---|---|---|---|---|---|---|---|
| LytR | Teichoic acid attachment protein LytR | SPD_1741 | 4.015 (3.488–4.455) | 0.022 (−0.087–0.084) | 0.149 (−0.003–0.235) | 0.074 (−0.052–0.080) | 0.217 (0.152–0.299) | Linear mixed effects |
| ManL | Glucose, mannose, galactose, fructose, N-acetylglucosamine and glucosamine ABC transporter ATPase ManL | SPD_0264 | 0.803 (0.374–1.553) | 0.000 (−0.104–0.059) | 0.460 (0.129–0.907) | 0.070 (0.052–0.469) | 1.083 (0.833–1.346) | Both |
| MltG | Endolytic murein transglycosylase MltG | SPD_1346 | 2.993 (2.330–3.371) | −0.087 (−0.119–0.055) | 0.249 (0.110–0.283) | 0.253 (0.060–0.401) | 0.491 (0.338–0.753) | Both |
| MreC | Peptidoglycan formation protein C MreC | SPD_2045 | 2.947 (1.440–4.467) | 0.133 (−0.048–0.199) | 0.092 (−0.021–0.219) | 0.123 (0.095–0.138) | 0.172 (0.076–0.344) | Linear mixed effects |
| Pbp1a | Penicillin-binding protein 1A | SPD_0336 | 1.543 (0.999–1.915) | −0.066 (−0.111–0.049) | 0.138 (0.016–0.256) | 0.082 (0.042–0.092) | 0.357 (0.192–0.734) | Both |
| Pbp1b | Penicillin-binding protein 1B | SPD_1925 | 0.615 (0.366–0.867) | −0.039 (−0.088–0.001) | 0.023 (−0.004–0.151) | 0.070 (−0.056–0.157) | 0.216 (0.108–0.412) | Empirical Bayes |
| Pbp2b | Penicillin-binding protein 2B | SPD_1486 | 4.348 (3.623–4.958) | −0.017 (−0.105–0.065) | 0.161 (0.037–0.293) | 0.426 (0.137–0.532) | 0.520 (0.189–0.770) | Both |
| Pbp2x | Penicillin-binding protein 2X | SPD_0306 | 2.934 (2.261–3.653) | 0.044 (−0.082–0.262) | 0.245 (0.012–0.426) | 0.174 (0.041–0.616) | 0.626 (0.315–1.174) | Both |
| Pbp3 | D-alanyl-D-alanine carboxypeptidase Pbp3 | SPD_0767 | 1.981 (1.253–3.408) | 0.006 (−0.031–0.036) | 0.053 (−0.030–0.137) | 0.042 (−0.020–0.448) | 0.209 (0.121–0.741) | Linear mixed effects |
| PclA | Pneumococcal collagen-like protein A | SPD_1376 | 0.638 (0.317–0.984) | −0.066 (−0.154 - −0.002) | 0.078 (−0.047–0.248) | 0.063 (0.007–0.163) | 0.177 (0.067–0.321) | Both |
| PgdA | Peptidoglycan-N-acetylglucosamine deacetylase PgdA | SPD_1309 | 2.407 (1.971–3.550) | −0.089 (−0.135–0.041) | 0.438 (0.154–0.812) | 0.111 (−0.004–0.286) | 0.511 (0.197–0.986) | Both |
| PiaA | Iron ABC transporter substrate-binding protein PiaA | SPD_0915 | 4.442 (4.142–4.795) | 0.001 (−0.076–0.086) | 0.086 (0.047–0.148) | 0.143 (0.127–0.145) | 0.287 (0.191–0.357) | Linear mixed effects |
| PiuA | Iron ABC transporter solute-binding protein PiuA | SPD_1652 | 3.593 (2.744–4.086) | 0.011 (−0.078–0.034) | 0.088 (−0.020–0.203) | 0.106 (0.091–0.167) | 0.410 (0.276–0.452) | Both |
| Ply | Pneumolysin | SPD_1726 | 1.306 (0.576–1.604) | −0.120 (−0.158 - −0.068) | 0.393 (0.205–0.428) | 0.338 (0.313–0.478) | 0.322 (0.162–0.799) | Both |
| PnrA | Ribonucleoside ABC transporter solute- binding protein | SPD_0739 | 4.634 (4.086–4.998) | −0.028 (−0.072–0.043) | 0.116 (0.047–0.203) | 0.084 (0.073–0.262) | 0.205 (0.112–0.464) | Linear mixed effects |
| PpmA | Foldase protein PpmA | SPD_0868 | 3.531 (2.883–4.059) | 0.079 (−0.078–0.112) | 0.412 (0.226–0.578) | 0.284 (0.264–0.825) | 0.594 (0.263–0.727) | Linear mixed effects |
| PppL | Protein phosphatase PhpP | SPD_1543 | 0.906 (0.711–1.867) | −0.066 (−0.253–0.005) | −0.066 (−0.195–0.189) | 0.021 (0.008–0.089) | 0.443 (0.055–0.630) | Empirical Bayes |
| PspA | Pneumococcal surface protein A | SPD_0126 | 3.616 (2.076–4.672) | −0.032 (−0.124–0.051) | 0.124 (0.023–0.241) | 0.144 (0.022–0.280) | 0.301 (0.103–0.484) | Both |
| PspC | Pneumococcal surface protein C | SPD_2017 | 2.810 (1.334–3.682) | −0.066 (−0.122–0.034) | 0.122 (−0.001–0.212) | 0.124 (0.060–0.188) | 0.285 (0.156–0.452) | Both |

*Table 1 continued on next page*

*Table 1 continued*

| Protein name | Functional annotation | *S. pneumoniae* D39 coding sequence | IgG binding at day 0 | $\Delta_{0->84}$ in placebo group | $\Delta_{0->84}$ in cohort 1 | $\Delta_{0->84}$ in cohort 2 | $\Delta_{0->84}$ in cohort 3 | Statistical analysis |
|---|---|---|---|---|---|---|---|---|
| Psr | Teichoic acid attachment protein Psr | SPD_1202 | 2.198 (1.514–2.946) | −0.028 (−0.081–0.069) | 0.161 (0.023–0.212) | 0.120 (0.089–0.333) | 0.328 (0.222–0.548) | Linear mixed effects |
| PstS2 | Phosphate ABC transporter solute-binding protein PstS2 | SPD_1232 | 3.479 (1.676–4.508) | 0.021 (−0.010–0.134) | 0.203 (−0.091–0.613) | −0.034 (−0.037 - −0.012) | 0.603 (0.314–0.945) | Linear mixed effects |
| PyrK | Dihydroorotate dehydrogenase B (NAD(+)) electron transfer subunit PyrK | SPD_0851 | 0.809 (0.643–0.991) | −0.087 (−0.139 - −0.020) | 0.032 (−0.080–0.058) | 0.014 (0.002–0.018) | 0.182 (0.143–0.225) | Empirical Bayes |
| QmcA | Membrane-associated protease regulator QmcA | SPD_1984 | 1.297 (0.852–1.812) | −0.041 (−0.094–0.013) | 1.144 (0.435–1.718) | 0.776 (0.643–1.311) | 1.249 (0.857–2.103) | Both |
| RexA | ATP-dependent helicase/nuclease subunit RexA | SPD_1016 | 0.443 (0.110–1.147) | −0.047 (−0.159–0.058) | 0.052 (−0.010–0.167) | 0.068 (0.024–0.073) | 0.401 (0.183–0.547) | Empirical Bayes |
| RmuC | DNA recombination protein RmuC | SPD_1778 | 2.496 (2.268–3.737) | −0.042 (−0.091–0.065) | 0.197 (0.080–0.241) | −0.006 (−0.041–0.092) | 0.253 (0.129–0.385) | Both |
| SepF | Cell division protein SepF | SPD_1477 | 0.427 (−0.263–1.073) | 0.079 (0.018–0.177) | 0.182 (0.033–0.646) | 0.083 (0.056–0.332) | 0.766 (0.324–1.050) | Both |
| SlrA or PpiA | Peptidyl-prolyl cis-trans isomerase SlrA or PpiA | SPD_0672 | 1.920 (1.103–3.274) | −0.006 (−0.061–0.091) | 0.551 (0.297–0.988) | 0.732 (0.691–0.858) | 1.360 (1.190–2.095) | Both |
| StkP | Serine/threonine-protein kinase StkP | SPD_1542 | 3.847 (3.282–4.111) | 0.047 (0.013–0.110) | 0.112 (0.096–0.124) | −0.004 (−0.076–0.087) | 0.178 (0.107–0.252) | Linear mixed effects |
| TagB | Membrane-associated protein TagB | SPD_1197 | 0.956 (0.797–1.621) | −0.038 (−0.075–0.075) | 0.074 (0.029–0.130) | 0.162 (0.145–0.163) | 0.124 (0.083–0.212) | Linear mixed effects |
| TcyA | ABC amino acid transporter solute-binding protein TcyA | SPD_0150 | 1.815 (1.279–2.538) | −0.096 (−0.160–0.055) | 1.117 (0.849–1.319) | 1.025 (0.324–1.724) | 0.947 (0.477–2.225) | Both |
| TprA | Quorum-sensing PclR-type transcriptional regulator TprA | SPD_1745 | 1.008 (0.654–1.284) | −0.054 (−0.099 - −0.039) | 0.029 (−0.092–0.095) | 0.056 (0.010–0.205) | 0.394 (0.237–0.474) | Both |
| Tuf | Elongation factor Tu | SPD_1318 | 0.306 (−0.003–1.146) | −0.065 (−0.323–0.006) | 0.777 (0.246–1.589) | −0.003 (−0.189–0.001) | 1.107 (0.481–1.483) | Both |
| YajC | Preprotein translocase YajC subunit | SPD_1838 | 1.580 (1.060–2.363) | −0.051 (−0.133–0.021) | 0.967 (0.494–2.145) | 1.454 (0.740–1.689) | 1.610 (0.977–2.225) | Both |
| YbbR | Uncharacterised protein YbbR | SPD_1391 | 3.584 (2.730–4.405) | −0.015 (−0.094–0.146) | 0.021 (−0.057–0.131) | −0.021 (−0.034–0.110) | 0.199 (0.060–0.540) | Linear mixed effects |
| YneF | Uncharacterised protein YneF | SPD_1662 | 0.187 (0.042–0.419) | −0.060 (−0.090–0.015) | 0.047 (−0.083–0.266) | 0.263 (0.133–0.299) | 0.314 (0.209–0.544) | Linear mixed effects |
| YoxC | Uncharacterised membrane protein YoxC | SPD_1242 | 0.160 (0.083–0.368) | 0.093 (0.041–0.170) | 0.107 (0.018–0.251) | 0.183 (0.123–0.226) | 0.410 (0.123–1.580) | Linear mixed effects |
| ZmpA | Zinc metalloprotease A | SPD_1018 | 3.084 (1.841–4.307) | −0.031 (−0.110–0.085) | 0.161 (0.043–0.288) | 0.142 (0.016–0.279) | 0.479 (0.260–0.700) | Both |
| ZmpB | Zinc metalloprotease B | SPD_0577 | 1.149 (0.436–2.096) | −0.042 (−0.152–0.032) | 0.127 (−0.031–0.206) | 0.089 (−0.037–0.202) | 0.139 (0.069–0.264) | Both |

DOI: https://doi.org/10.7554/eLife.37015.028

(*Giefing et al., 2008*), but the consistent rise in IgG binding observed in this trial corresponded to a small increase on a low baseline.

Excluding the CBD oligomer and grouping together orthologous variants, the 112 WCV antigens identified by the eBayes and LMM analyses were found to correspond to 72 functionally distinct proteins, of which all but four could be attributed to sequences in the *S. pneumoniae* RM200 genome (*Supplementary file 1 and 2*). A multivariable analysis of the functional characteristics distinguishing the immunogenic proteins from those in the WCV not provoking an elevated IgG response was used to test whether these antigens were enriched for particular functional or structural characteristics. This did not identify CBDs as a marker of antigenicity (*Supplementary file 3*), suggesting CBD-binding IgG did not elevate the overall antibody response to all proteins containing this motif. Nevertheless, such antibodies could potentially explain one inconsistency, in which the LMM identified elevated IgG binding to the lytic amidase LytA, despite only a 15 aa N-terminal fragment remaining in the RM200 strain used for immunization (*Figure 1—figure supplement 1*). This could be attributed to the multiple CBDs of LytA on the array being the epitope recognised by vaccine-induced antibodies. Another protein associated with an elevated IgG response, pneumolysin (CLS01670), seems to have caused a measureable response despite small modifications to remove its cytolytic activity in the WCV (*Lu et al., 2010b*).

The multivariable analysis of protein characteristics found a significant association between increased IgG binding and signal peptides, which direct proteins for export across the cell membrane (odds ratio = 6.76, p = $1.38 \times 10^{-5}$; *Supplementary file 3*). This corresponded with the strong immune responses to many surface-associated proteins. Excepting the transcriptional regulator, FrlR (CLS00137), the greatest $\Delta_{0 \to 84}$ values were measured for three conserved surface-exposed proteins: a pre-protein translocase YajC (CLS01753), a protease regulator HflC (CLS01867), and the peptidyl-prolyl isomerase PpiA (CLS00702), the only one of these four proteins to register as an ABT in the pre-vaccination sample (*Figure 4B*). Another ABT involved in maintaining surface proteins' conformation, the foldase PpmA (CLS00885), was identified as eliciting an IgG response by the LMM analysis.

Other protein motifs showing similarly strong associations in the multivariable analysis included the Transpeptidase domain, associated with peptidoglycan remodelling, and the SBP_bac_3 domain, associated with ATP-binding cassette (ABC) transporter solute-binding proteins (SBPs), which bind exogenous substrates and deliver them to cognate permeases for import into the cell. Accordingly, a broad functional categorisation found cell wall metabolism proteins and SBPs were similarly represented in the WCV antigens as in pre-vaccination ABTs (*Croucher et al., 2017*), whereas relatively few vaccine-induced responses were observed to adhesins and surface-associated degradative enzymes (*Figure 4C*).

## WCV-induced responses to transporter proteins

Of the 27 SBPs in the RM200 genome, 25 were among the 1062 proteins that were immunoreactive and highly similar between the WCV strain and the panproteome array. Eleven of the 25 (44.0%) provoked a substantial increase in IgG binding, relative to 60 of the 1037 non-SBPs, an 7.60-fold enrichment representing their significant contribution to the WCV response (Fisher's exact test, $N = 1,062$, p = $1.16 \times 10^{-7}$). SBPs were enriched in the subset of ABTs triggering significant further IgG rises, accounting for eleven of the 71 ABTs (15.5%; a 3.54-fold enrichment) identified as immunogenic by the eBayes or LMM analyses, but only six of the 137 ABTs (4.38%) not associated with a significant increase (Fisher's exact test, $N = 208$, p = 0.014). Two of the ABT SBPs triggering a post-WCV rise in IgG binding were the siderophore transporters PiaA and PiuA, which have been considered as potential protein vaccine candidates (*Jomaa et al., 2006*). Seven other immunogenic SBPs bound amino acids or peptides for import by ABC transporters. These included a large increase in IgG recognising TcyA (CLS00206; *Figure 4B*), and rises to the glutamine-binding proteins GlnH (CLS01210), GlnPH4 (CLS01088) and GlnPH1 (CLS00459), only the former two of which were classified as ABTs in the pre-vaccination data. Nevertheless, other SBPs strongly recognised by natural immunity showed no sign of increased IgG binding following vaccination (*Figure 4—figure supplement 8*).

Other transporter proteins eliciting a significant elevation in IgG binding were FruA (CLS00796), the substrate-recognising IIC subunit of a fructose-specific phosphotransferase system importer, and protein CLS02831 (*Supplementary file 2*), a bacteriocin exporter present in RM200 but only found in 29% of the pneumococcal population of Massachusetts (*Croucher et al., 2013a*). Another

bacteriocin protein to induce elevated IgG binding was the secreted CibA competence-induced bacteriocin, suggesting at least some of the pneumococci were in the 'X state' when killed prior to inoculation (*Claverys and Håvarstein, 2007*).

## WCV-induced responses to cell wall metabolism proteins

Sixteen of the 25 previously-defined ABTs (64.0%) involved in cell wall metabolism were among the 71 ABTs provoking a significant increase in IgG binding by at least one of the statistical analyses (*Figure 4—figure supplement 9*), corresponding to a 3.43-fold enrichment (Fisher's exact test, $N = 208$, p = 0.0014). Additionally, four functionally similar proteins that were not classified as ABTs on the basis of pre-vaccination immune responses elicited significantly elevated IgG binding (*Supplementary file 2*). As well as Pbp1B (CLS01817), these included variants of the Pbp1A protein, which, along with Pbp2B and Pbp2X, have diverged under selection for resistance to β-lactams (*Croucher et al., 2013a*). The penicillin-binding protein sequences in RM200 are most similar to the ancestral, β-lactam sensitive variants (*Figure 4—figure supplement 10*). While there were subtle differences in the immune response to different variants of all three of these proteins, both the dose response and maximum increase in IgG binding were similar for each (*Figure 4—figure supplement 11*).

## Specificity of WCV antibody responses to polymorphic proteins

Loci exhibiting higher levels of divergence correspondingly provided greater evidence of variant-specific immune responses. Two of the adhesins to which elevated IgG binding was detected were variants of the PclA protein (*Paterson et al., 2008*) (*Figure 4—figure supplement 12* and *Supplementary file 2*). These were the most similar functional variants to the PclA sequence in the *S. pneumoniae* RM200 genome (*Figure 4—figure supplement 13*), with a truncated version (CLS99466) and two divergent full-length proteins (CLS03178 and CLS03616) not showing similar increases (*Figure 4—figure supplement 14*). No increased IgG binding to pilus subunits or serine-rich glycoproteins was observed, in keeping with their absence from the RM200 genome (*Figure 4—figure supplement 14*). These data are consistent with both allelic divergence, and absence of antigens encoded by genomic islands, being an effective means of pneumococcal strains evading adaptive immune responses induced by colonisation with genetically-distinct strains.

The proteins represented by the greatest number of variants on the array were pneumococcal surface proteins A and C, encoded by the DCL *pspA* and *pspC*. The array contained no variant highly similar to RM200's PspA or PspC sequences. Nevertheless, a subset of the variants of each showed a significant increase in IgG binding that was most pronounced in cohort 3, and absent from the placebo group (*Figure 4B* and *Figure 5A,B*). These changes are unlikely to be non-specific responses caused by IgG recognising the CBDs of these proteins, as the variants identified as immunogenic contained a lower mean number of CBDs (5.58 for PspA, 6.00 for PspC) than the average number of such domains across all variants on the array (6.97 for PspA, 6.17 for PspC). However, this induced response was also not simply related to sequence divergence across the rest of the protein. For PspA, IgG binding rose against multiple variants, consistent with previous data suggesting broadly protective anti-PspA immunity can be induced by immunisation with a single variant, despite the diversity of this protein (*Nabors et al., 2000*; *Tart et al., 1996*). By contrast, the response to PspC was only strongly evident to variant 42, which had an intermediate level of sequence similarity to the PspC of RM200 compared to other variants on the array. This observation could not be attributed to cross-reactivity with PspA (*Brooks-Walter et al., 1999*), as this PspC allele is not closely related to the WCV PspA. Hence this may represent a relatively poor immune response to PspC, owing to its weakened association with the cell surface following the loss of some of its CBDs, that triggers recognition of a specific epitope unique to variant 42.

Surface-exposed degradative enzymes generally elicited little increase in IgG binding (*Figure 4—figure supplement 15*). Two exceptions were proteins ubiquitous across encapsulated pneumococcal lineages (*Croucher et al., 2014*): the β-galactosidase BgaA (CLS00596) and serine protease HtrA (CLS00066) (*Croucher et al., 2017*) (*Figure 4—figure supplement 15*). There was also a significant response to the zinc metalloproteases encoded by DCL, ZmpA and ZmpB (*Figure 4B* and *Figure 5C,D*). The eBayes analysis only identified a significant IgG binding increase to a single ZmpB variant (*Supplementary file 2*), which was the sequence most closely related to that found in

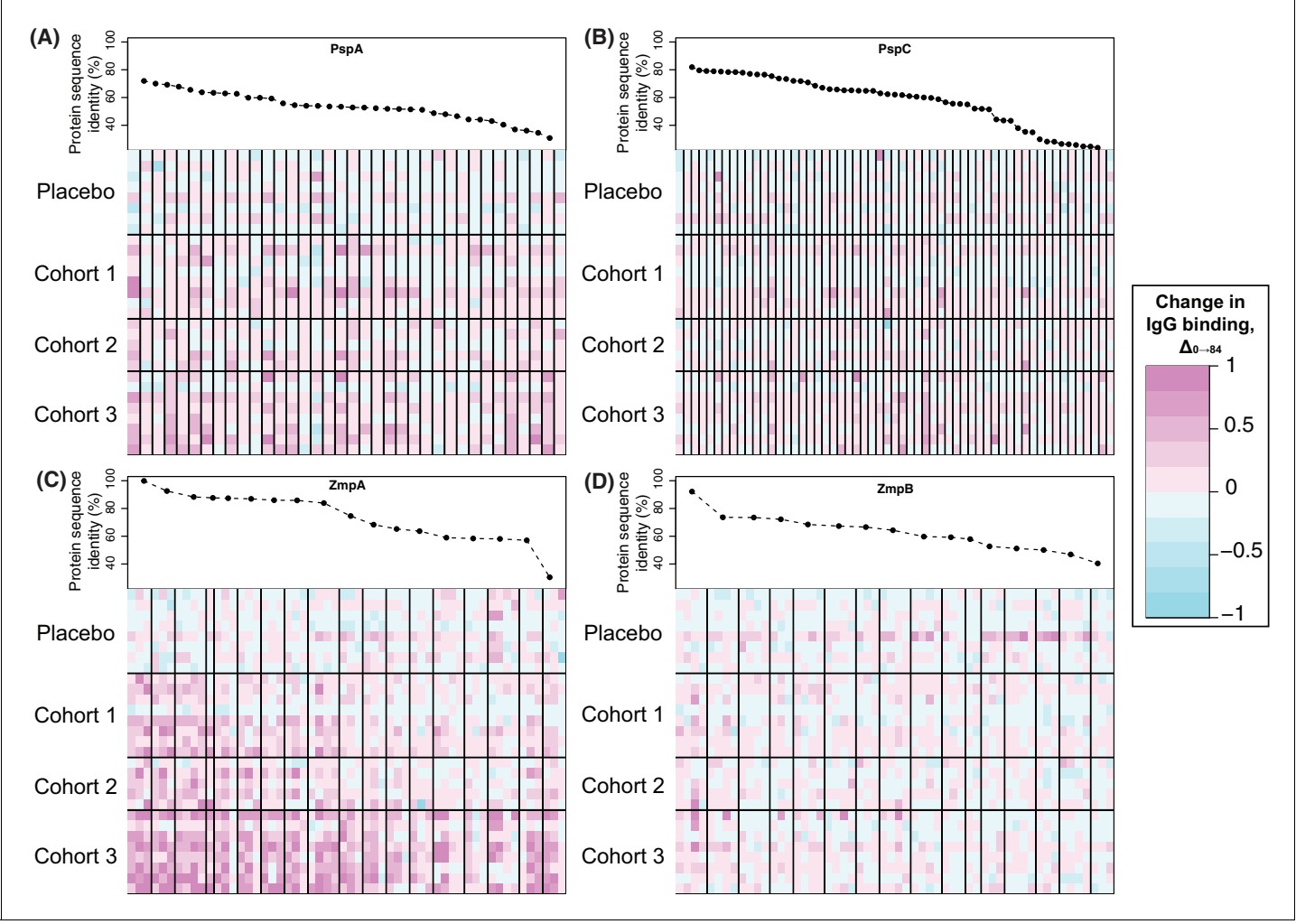

**Figure 5.** Changes in IgG binding to diverse core loci. Each heatmap has a row for each of the 29 trial participants with samples at the start and end of the trial, ordered by cohort. Each column represents a probe on the proteome array, with vertical black bars separating the probes from different variants. The cells are coloured by the direction and magnitude of $\Delta_{0\to84}$, with increasingly strong cyan representing reduced IgG binding over the duration of the trial, and increasingly strong magenta representing elevated IgG binding. The variants are ordered according to their similarity to the corresponding sequence in the WCV, with the most similar on the left, and least similar on the right. (**A**) Heatmap showing $\Delta_{0\to84}$ for pneumococcal surface protein A (PspA) variants. (**B**) Heatmap showing $\Delta_{0\to84}$ for pneumococcal surface protein C (PspC) variants. (**C**) Heatmap showing $\Delta_{0\to84}$ for zinc metalloprotease A (ZmpA) variants. The large zinc metalloprotease proteins are typically represented by multiple probes. (**D**) Heatmap showing $\Delta_{0\to84}$ for zinc metalloprotease B (ZmpB) variants.

DOI: https://doi.org/10.7554/eLife.37015.029

RM200; all other alleles exhibited >20% sequence divergence, suggesting a variant-specific immune response. By contrast, IgG binding to many ZmpA variants was increased by WCV administration (*Figure 4B and 5C*). Consistent with the results with ZmpB, however, the strongest responses were generally associated with those variants with <20% protein sequence divergence with RM200's ZmpA, again suggesting a simpler relationship between antibody cross-reactivity and immune responses than for PspA and PspC (*Figure 5*). Nevertheless, there were some highly divergent ZmpA variants associated with elevated IgG binding post-vaccination, demonstrating some epitopes had a complex distribution across the population.

## Differing kinetics of WCV responses between antigens

Plotting the changes in IgG binding between each sampling point for proteins found to elicit significantly elevated responses by either the LMM or eBayes analyses showed differences in the timing of

responses between functionally-defined categories (*Figure 6*). For SBPs, cell wall metabolism proteins, and ZmpA variants, responses were generally maximal after a single injection, particularly for the cohort receiving the highest dose of WCV. The empirically-determined threshold of 0.2 was used to identify substantial changes in IgG binding between consecutive timepoints among the probes to which there was a significant overall vaccine response (*Figure 4—figure supplement 2*). One hundred and twelve probes increased by more than this cutoff in the 28 days after the first vaccine dose (*Figure 7*). These probes had a high median pre-vaccination IgG binding of 2.53, and likely represent many cases of anamnestic responses being triggered.

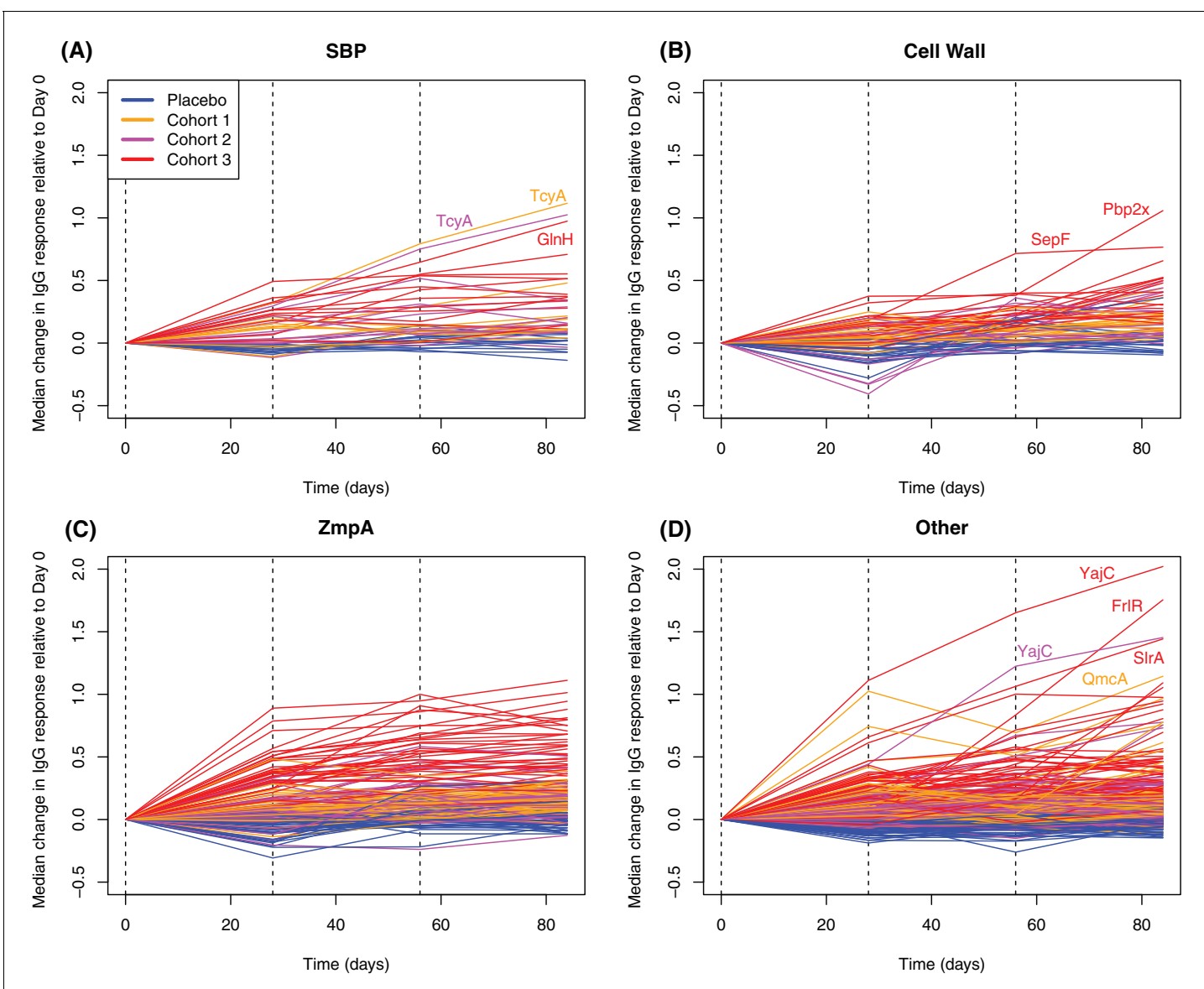

**Figure 6.** Longitudinal responses to successive WCV doses. These plots show the within-cohort median change in IgG binding, relative to the level in the pre-vaccination sample, using data from the 27 individuals with samples taken at all four timepoints. The probes are functionally grouped as described in *Supplementary file 2*: (A) solute-binding proteins; (B) proteins involved in cell wall metabolism; (C) zinc metalloprotease A variants; and (D) other core proteins. Only probes found to have a significant increase in IgG binding after WCV administration by eBayes or LMM analyses are included. Between each sampled timepoint, the solid lines join the median IgG-binding values for the cohort indicated by their colour. Vertical dashed lines show times of WCV administration.
DOI: https://doi.org/10.7554/eLife.37015.030

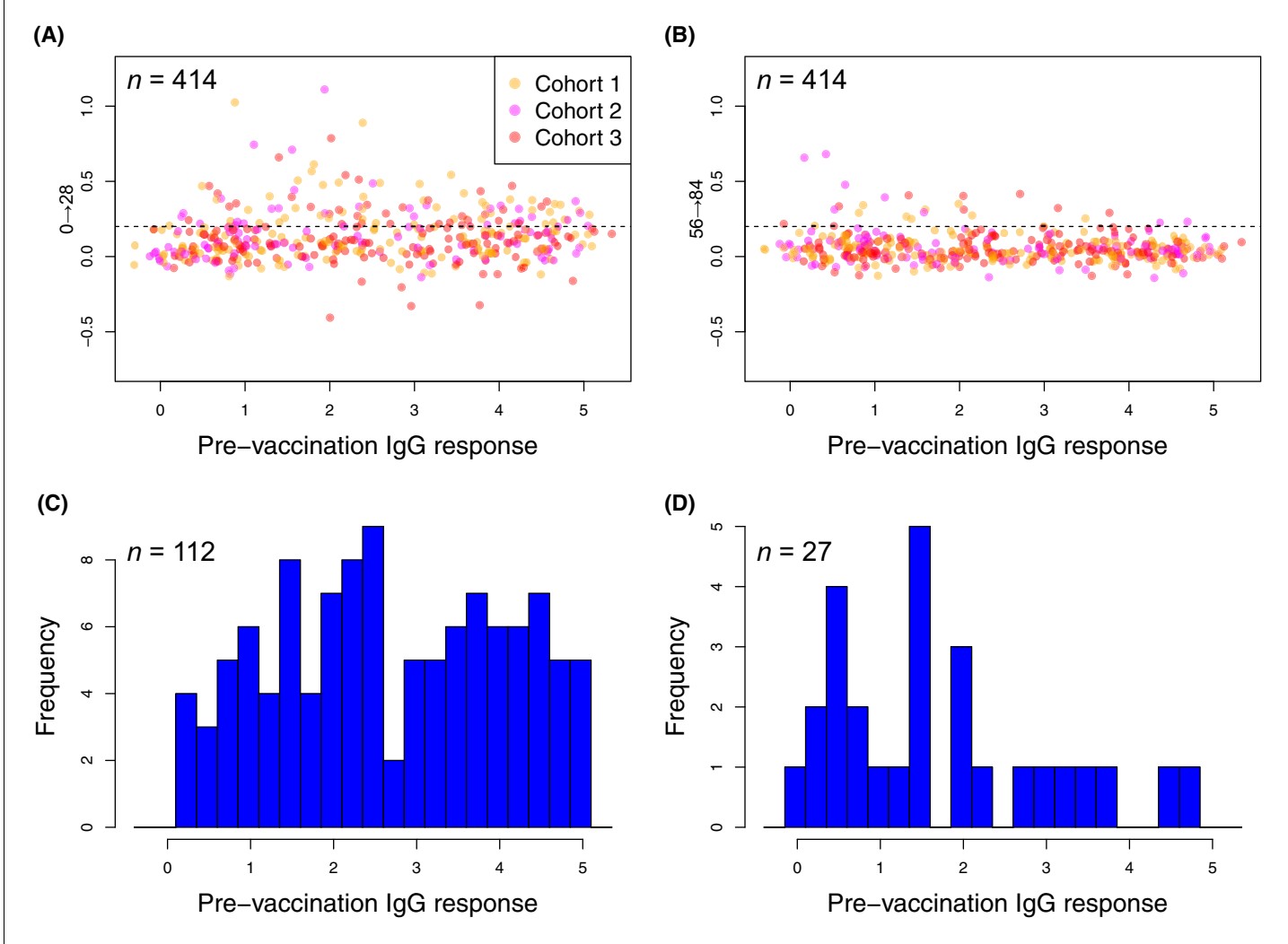

**Figure 7.** Changes in the within-cohort median IgG binding between consecutive timepoints. (**A and B**) The scatterplots show 414 points, corresponding to the 138 probes found to have a significant increase in post-WCV IgG binding by the eBayes or LMM analyses in each of the three vaccinated cohorts, as no induced immunity is expected in the placebo group. The horizontal dashed line represents the empirically-determined threshold of 0.2 used for identifying those probes associated with an atypically large change in IgG binding between successive timepoints. Points are coloured according to the cohort from which they derive, as indicated by the key. Panel (**A**) shows the within-cohort median changes in IgG binding following the first inoculation ($\Delta_{0 \to 28}$) relative to the within-cohort median pre-vaccination level of IgG binding. Panel (**B**) shows the equivalent plot following the third inoculation ($\Delta_{56 \to 84}$). (**C and D**) For the responses (defined by probe and cohort) exceeding the specified threshold, the corresponding pre-vaccination IgG-binding values are shown by the histograms of within-cohort median IgG binding at day 0. Panel (**C**) shows the 112 high responses to the first WCV dose ($\Delta_{0 \to 28} > 0.2$) span the full range of pre-vaccine IgG-binding values, whereas panel (**D**) shows the 27 high responses to the third WCV dose ($\Delta_{56 \to 84} > 0.2$) are typically associated with low pre-vaccine IgG binding.
DOI: https://doi.org/10.7554/eLife.37015.031

In contrast, only 27 probes detected a change of similar magnitude after the third WCV dose (*Figure 7*). These were associated with comparatively low pre-WCV antibody responses. The pre-vaccination IgG binding to these probes, and to the 112 probes to which a substantial rise was detected after the first dose, was compared using a Wilcoxon rank sum test. This found these late-responding probes to have a significantly lower level of pre-vaccination IgG binding (median 1.43; $N = 139$, $W = 2116.5$, $p = 0.0013$). These kinetics probably represent the slower development of a primary immune response to proteins not eliciting antibodies prior to WCV administration. Such reactivity was typically associated with the set of proteins with diverse functional annotations (*Supplementary file 2*), not falling into the main functional categories associated with natural

immunity. The largest changes were for the pre-protein translocase subunit YajC and the intracellular transcriptional regulator FrlR, neither of which showed high levels of IgG binding in the pre-vaccination samples (*Supplementary file 2*; *Figure 6*), as well as the surface-exposed peptidyl-prolyl isomerase PpiA. Hence multiple doses of the WCV can elicit antibody responses to proteins not commonly recognised by pre-existing immunity.

## Discussion

This panproteome-wide analysis of the changes in IgG binding following administration of a WCV provides a comprehensive view of the antibodies generated in response to systemic inoculation with an unencapsulated pneumococcus. The phase I clinical trial of WCV was primarily designed to measure safety outcomes, thus statistical power to assess immunogenicity at the proteome scale may have been limited. Nevertheless, complementary statistical analyses of changes in antibody response found significantly elevated IgG binding to 137 probes, corresponding to 112 protein sequences. Accounting for identification of orthologous variants, these responses spanned 72 functionally-distinct proteins, including multiple conserved antigens. These changes in immunity appear to represent a combination of boosting anamnestic responses to previously-recognised antigens, as well as the induction of novel antibodies to previously-unrecognised antigens. This complexity likely at least partially reflects the participants in this trial being healthy adults, with mature pre-existing immunity to pneumococci that is likely to limit the WCV-induced IgG response. Administration of WCV to more immunologically-naïve infants, the intended recipients of the vaccine, would likely result in a much higher proportion of novel antibody responses.

The distinctive antibody fingerprints of the adults in the trial suggest humans' mature antibody repertoire is shaped by multiple encounters with diverse pneumococci, as the overall modifications caused by three WCV inoculations were relatively small. However, these alterations were remarkably consistent between individuals, despite the pre-existing differences in their adaptive immune responses. Only a subset of proteins, enriched for naturally immunogenic ABTs, provoked a strong post-WCV response. The reasons for this heterogeneity are unclear. These differences in $\Delta_{0\rightarrow84}$ measurements were evident even between proteins with similar functional and structural characteristics (*Figure 4—figure supplement 8* & *Figure 4—figure supplement 9*), or of similar immunogenicities, based on the pre-WCV samples (*Figure 3*). This pattern is unlikely to represent a lack of sensitivity due to saturation of the system, occurring at IgG-binding levels above six in this study, which were not frequently reached in these data. Nor is it likely attributable to sequence divergence between RM200 and protein representatives on the array, given probes' high sequence similarity to some ABTs encoded in the RM200 genome that failed to elicit a substantial IgG response (*Supplementary file 1*).

This suggests relatively subtle differences in protein characteristics, or antibody response kinetics, could affect which proteins are recognised most strongly when presented in the WCV. Alternatively, it may relate to differences in protein concentrations between the WCV and live pneumococci encountered by the host. However, it would seem unlikely that chemical killing, and subsequent processing, of the RM200 cells would retain a subset of each of the immunogenic SBPs, attached to the membrane through a lipid moiety, as well as surface proteins PspA and PspC, attached to cell wall polysaccharide through choline-binding domains, and ZmpA, covalently attached to the cell wall via sortase activity. It is more conceivable that many ABTs may not have been expressed by the RM200 cells during culture, although the strong and apparently novel response to some predominantly intracellular proteins, such as FlrR, suggests expression of surface-associated proteins would have to be very low for lack of exposure to explain the absence of an immune response.

The surface-associated proteins enriched among WCV antigens may be important components of a protective IgG response, as recognition of these structures enables intact *S. pneumoniae* cells to be agglutinated (*Mitsi et al., 2017*) or targeted for opsonophagocytosis (*Hyams et al., 2010*). However, the high-throughput nature of the array measurements of antibody binding cannot currently substitute for functional assays in determining whether these IgG responses are sufficient for protection. It is not clear whether a threshold correlate of protection, as determined for anti-capsular antibodies induced by PCVs (*Andrews et al., 2014*), could be applicable to a broad range of distinct protein antigens. One potentially confounding factor would be the accessibility of the different surface antigens on encapsulated strains, which suggests another explanation for the pattern of

immune responses. Antibody binding of SBPs and components of the cell wall synthesis machinery is impeded by the capsule of most *S. pneumoniae* (*Gor et al., 2005*). These proteins accounted for much of the increased IgG binding following administration of the WCV. By contrast, adhesins and surface-associated degradative enzymes must extend beyond the capsule to function, and were comparatively absent from the antigens triggering post-WCV increases in IgG binding. Hence the observed enrichment of the former functional categories in the immune response to the vaccine could represent IgG recognition of the latter, larger surface proteins already being maximal in healthy adults, a situation that would be less likely in children receiving the same vaccine.

The opportunity for the adaptive immune system to strengthen responses to SBPs and the cell wall machinery may be particularly valuable, as many of these antigens are conserved across the population (*Croucher et al., 2017*). Other subcapsular, conserved surface ABTs, such as the PpmA and PpiA foldases, were also associated with strong IgG responses. This is consistent with previous work with sera from animals inoculated with the RM200 WCV showing antibody-binding pneumococci of different serotypes (*Campos et al., 2017*; *Gonçalves et al., 2014*). Therefore, these antibodies could afford broad protection against a diverse set of strains, as long as these antibodies can penetrate the capsular envelope to bind their cognate proteins. They would likely be more effective against transparent phase variants, in which lower levels of capsule are expressed (*Weiser et al., 1994*). These are more commonly associated with colonising isolates adhering to the nasopharyngeal epithelium, rather than those causing disease.

The few strong responses that were observed to adhesins, such as PclA and PspA, and degradative enzymes, such as ZmpA, may reflect exposure to new variants of diverse proteins that an individual has not encountered previously. In each of these cases, there was evidence of some variant-specificity in the response, although this was not simply related to sequence similarity between the proteins in the WCV and the variants on the array, suggesting these proteins contain multiple epitopes with a complex distribution across orthologues. For ZmpA, ZmpB and PclA, heterogeneity in the pattern of IgG binding suggests the evident sequence divergence is enough to at least weaken, if not completely evade, IgG binding induced by other variants, consistent with adaptive immunity being an important driver of the divergence of these antigenic loci (*Croucher et al., 2017*; *Li et al., 2012*). The contrasting binding of multiple PspA variants by WCV-induced antibodies could be an artefact of the absence of a variant on the array that was closely-related to the PspA of RM200, such that the measured responses are all similarly high because all representatives on the array are relatively distant from the variant that stimulated the response. Some previous work has found antibodies induced by individual PspA or PspC representatives to be broadly protective against a wide spectrum of variants (*Brooks-Walter et al., 1999*; *Nabors et al., 2000*; *Tart et al., 1996*), consistent with induction of genuinely cross-reactive IgG, although other studies have found evidence for greater specificity in anti-PspC responses (*Georgieva et al., 2018*). Therefore, inducing antibodies capable of recognising multiple variants of DCL could contribute to the protection induced by this WCV formulation being effective across the species.

Although this study focuses on the humoral response to WCV, cellular responses mediated by CD4[+] T cells and interleukin 17A are important in the vaccine's role in inhibiting nasopharyngeal colonisation in a mouse model (*Campos et al., 2017*; *Lu et al., 2010a*). Nevertheless, in the same animal model, WCV-induced protection against invasive disease was more dependent on protective antibodies than cellular immunity (*Lu et al., 2010a*). In humans, individuals with agammaglobulinemia, who have low levels of circulating antibodies, are highly susceptible to pneumococcal disease, atesting to the importance of protective antibody-mediated humoral immunity (*Conley et al., 2000*). While anticapsular antibodies are important in protecting against pneumococcal colonisation and disease, there is evidence that antibodies recognising proteins may be effective in these roles as well, based on their diversification that indicates immune selection (*Croucher et al., 2017*; *Li et al., 2012*), the effects of protein vaccines in various animal models (*Briles et al., 2000*; *Gor et al., 2005*; *Jomaa et al., 2005*), and the clinical effectiveness of protein-binding antibodies in intravenous immunoglobulin preparations used to treat pneumococcal disease (*Wilson et al., 2017*).

These data provide new information on how this critically important pneumococcal antibody repertoire develops, both specifically relating to systemic immunisation with WCV, but likely also serving as a model of how humoral immunity responds to natural exposure to pneumococci. Such routine contacts with the bacterium are common and likely to have been somewhat confounding in this study, as the data in *Figure 5* suggest some individuals in the placebo group may have encountered

a naturally-circulating pneumococcus with different DCL alleles, and thus mounted a different pattern of IgG responses to the panel of variants. Despite such strain-specificity, there are many strong responses to the WCV that are consistent across individuals, congruent with the generally similar patterns of pre-vaccination immunity in this study (*Croucher et al., 2017*) and the comparable profiles evident in pooled immunoglobulins collected from different countries (*Wilson et al., 2017*). This suggests the development of adaptive immunity is similar between individuals, and therefore these data are not just informative about the mechanism by which WCV may provide protection against pneumococcal disease, but also help build a more general understanding of the development of the natural immune repertoire, and what consequences this has for pneumococcal biology.

# Materials and methods

## Key resources table

| Reagent type (species) or resource | Designation | Source or reference | Identifiers | Additional information |
|---|---|---|---|---|
| strain | *Streptococcus pneumoniae* RM200 | PMID:20427625 | ENA:SAMEA104558347 | Dr. Rick Malley (Boston Children's Hospital) |
| commercial assay | *Streptococcus pneumoniae* Pan-Genome Microarray | Antigen Discovery Inc. | | Dr. Joseph Campo (Antigen Discovery Inc.) |
| software | R | https://www.r-project.org/ | | |

Information on the samples has been provided as *Figure 1—source data 1*, and the standardized IgG-binding values from the panproteome array have been provided as *Figure 2—source data 1*.

### Genomic analyses

A draft genome of *S. pneumoniae* RM200 was generated through a combined assembly of 454 and Illumina sequencing data using Celera Assembler v6.1 (*Myers et al., 2000*). These contigs were aligned to the *S. pneumoniae* D39 reference genome (accession code: CP000410) (*Lanie et al., 2007*), and one cut at the origin of replication, using ACT v13.0.0 (*Carver et al., 2008*). The resulting annotated assembly has been deposited in the European Nucleotide Archive with the sample accession code ERS2169631. The genomes of D39 and RM200 were then aligned using MAFFT v7.221 with default settings (*Katoh and Standley, 2013*). Base substitutions and recombinations distinguishing the two sequences were identified with Gubbins v1.4.10 (*Croucher et al., 2015a*). Pairwise comparisons with *S. pneumoniae* D39 and OXC141 (*Croucher et al., 2013b*) (accession code: FQ312027) were performed with BLASTN (*Camacho et al., 2009*) and ACT.

The COGsoft package was used to link the proteome of individual genomes to the probes on the array (*Kristensen et al., 2010*). The coding sequences of *S. pneumoniae* RM200, *S. pseudopneumoniae* IS7493 (accession code: CP002925), *S. mitis* B6 (accession code: FN568063) and *Streptococcus mutans* UA159 (accession code: AE014133) were identified using the methods described previously (*Corander et al., 2017*). The proteins were then aligned to those from a collection of 616 genomes from Massachusetts (*Croucher et al., 2015b*), used to design the proteome array (*Croucher et al., 2017*), using BLASTP (*Kent, 2002*). These comparisons were then processed using COGcognitor (*Kristensen et al., 2010*). For each of the probes on the array, every protein matching the same COG in a given genome was aligned to the protein used to design the probe using MAFFT (*Katoh and Standley, 2013*), and the maximal sequence identity recorded (*Supplementary file 1*). *S. mutans* proteins were sufficiently divergent from those in *S. pneumoniae* to suggest conservation should be defined only using the mitis group species, *S. mitis* and *S. pseudopneumoniae*. An analogous COGsoft analysis was also used to link the proteins on the array to the genome of *S. pneumoniae* D39 to add the functional annotation to *Table 1* (*Lanie et al., 2007*).

Solute-binding proteins were identified by using hmmscan v3.1 (*Eddy, 2011*) to search the *S. pneumoniae* RM200 proteome using the relevant Pfam domains, identified by keyword searches: SBP_bac_1 (accession code PF01547.24), SBP_bac_3 (accession code PF00497.19), SBP_bac_5 (accession code PF00496.21), SBP_bac_6 (accession code PF13343.5), SBP_bac_8 (accession code PF13416.5), SBP_bac_10 (accession code PF07596.10), SBP_bac_11 (accession code PF13531.5), Peripla_BP_1 (accession code PF00532.20), Peripla_BP_2 (accession code PF01497.17), Peripla_BP_3

(accession code PF13377.5), Peripla_BP_4 (accession code PF13407.5), Peripla_BP_5 (accession code PF13433.5), Peripla_BP_6 (accession code PF13458.5), ABC_sub_bind (accession code PF04392.11), Bmp (accession code PF02608.13), DctP (accession code PF03480.12) and ZnuA (accession code PF01297.16). AliA was not identified by this analysis, but was included as an SBP as it had been identified as such in a previous analysis (*Croucher et al., 2017*).

For the PspA, PspC, ZmpA and ZmpB variants, sequence identity values were calculated by aligning the full-length proteins on the proteome array with the corresponding allele in RM200 using MAFFT (*Katoh and Standley, 2013*). For the penicillin-binding protein and PclA variants, the proteins were again aligned with MAFFT, and the phylogenies generated with FastTree2 (*Price et al., 2010*).

## Study design and participants

The VAC-002 phase one study (ClinicalTrials.gov identifier: NCT01537185) was approved by the Western Institutional Review Board and conducted in compliance with the study protocol, international standards of Good Clinical Practice and the Declaration of Helsinki. Participants were healthy adults aged 18 to 40 years at the time of consent, and had no evidence of chronic health issues, nor any history of invasive pneumococcal disease or pneumococcal vaccination. Forty-two participants were enrolled and either received a WCV dose or a saline placebo, with sequential subject assignment performed by data management using an electronic randomization block design. Pharmacy staff responsible for preparing and administering vaccinations were unblinded. All others involved in conducting the trial, including participants, remained blinded to treatment assignment.

This study is a *post hoc* analysis of the samples available at the end of the clinical trial, and does not present the original safety and immunogenicity tests for which the trial was designed. No specific power analysis was conducted prior to the trial, or the sample selection for this subsequent study.

## Proteome microarray analysis of serum samples

The *Streptococcus pneumoniae* Pan-Genome Microarray, produced by Antigen Discovery, Inc. (ADI, Irvine, CA, USA), was designed and assayed as described previously (*Croucher et al., 2017*). Briefly, the microarray included 4504 full-length or fragmented proteins from the TIGR4 reference genome (2106 genes) and representatives of 2190 proteins identified in the Massachusetts pneumococcal population (*Croucher et al., 2013b*). These correspond to 2055 clusters of orthologous genes (COGs), 36 PspA variants, 57 PspC variants, 18 ZmpA variants, 16 ZmpB variants, and individual sequences for LytA, a phage amidase, ZmpE, PblB, PsrP and a choline-binding domain oligomer. Proteome microarrays were fabricated using a library of partial or complete CDSs cloned into the T7 expression vector pXI. Proteins were expressed using an *E. coli in vitro* transcription and translation (IVTT) system (Rapid Translation System, 5 Prime, Gaithersburg, MD, USA) and printed onto nitrocellulose-coated glass AVID slides (Grace Bio-Labs, Inc., Bend, OR, USA) using an Omni Grid Accent robotic microarray printer (Digilabs, Inc., Marlborough, MA, USA). Microarrays were probed with sera and antibody binding detected by incubation with biotin-conjugated goat anti-human IgG (Jackson ImmunoResearch, West Grove, PA, USA), followed by incubation with streptavidin-conjugated SureLight P-3 (Columbia Biosciences, Frederick, MD, USA). Slides were scanned on a GenePix 4300A High-Resolution Microarray Scanner (Molecular Devices, Sunnyvale, CA, USA), and raw spot and local background fluorescence intensities, spot annotations and sample phenotypes were imported and merged in R (*R Core Team, 2017*), in which all subsequent procedures were performed. Foreground spot intensities were adjusted by local background by subtraction, and negative values were converted to one. All foreground values were transformed using the base two logarithm. The dataset was normalised to remove systematic effects by subtracting the median signal intensity of the IVTT controls for each sample. With the normalised data, a value of 0.0 means that the intensity is no different than the background, and a value of 1.0 indicates a doubling with respect to background. Values below −2 in the normalised data, corresponding to less than 0.25 of the IVTT control probe signals, were adjusted to −2. This affected 152 of the 304,590 binding values included in the study, of which a further five were missing. Immunoreactivity was defined as achieving an IgG-binding level of one or greater in the study population at any timepoint being analysed. Change in

antibody levels ($\Delta_{0 \to 84}$) was calculated as IgG-binding levels from post-vaccination (day 84) minus pre-vaccination (day 0) IgG-binding levels.

## Statistical analysis of IgG binding

The t-SNE analyses were calculated after 25,000 iterations with a perplexity parameter of 40 using the R package Rtsne (*Krijthe, 2015*). Empirical Bayes (eBayes) analyses were conducted with the R package limma (*Ritchie et al., 2015*). The eBayes analysis was designed as pairwise contrasts of $\Delta_{0 \to 84}$ values from each of the vaccinated cohorts against those from the placebo group, with no intercept in the model. Only probes that exhibited immunoreactivity, as defined previously, were used to avoid reporting low level changes in IgG binding that would likely not be of biological relevance. Only the 29 trial participants with samples at both day zero and day 84 were included in this analysis. The 1584 p values associated with all immunoreactive probes, or the 1384 non-DCL immunoreactive probes, were subject to a Benjamini-Hochberg correction (*Benjamini and Hochberg, 1995*) using the limma function topTable, and reported as significant if they were below the false discovery rate threshold of 0.05.

The linear mixed effects models were fitted using the R package lme4 (*Bates et al., 2015*). To test for the relationship between cohort and $\Delta_{0 \to 84}$, a model was fitted to all probes, or immunoreactive probes only, using the $\Delta_{0 \to 84}$ from the 29 trial participants with samples at both day zero and day 84. The model had the form:

$$\Delta_{0 \to 84, p} = X_c c + X_p p + Z_S + \varepsilon$$

Where $\Delta_{0 \to 84, p}$ is the change in IgG binding to a probe $p$ over the duration of the trial; $X_c$ is the fixed effect of an individual's cohort, $c$; $X_p$ is the fixed effect of measuring IgG binding using probe $p$; $Z_s$ is the random effect associated with each trial participant $s$, and $\varepsilon$ is an error parameter. The likelihood of $X_c$ being non-zero was calculated through fitting an identical model without the $X_c c$ term, and comparing the likelihood ratio of the two models with an ANOVA test.

The per-probe model fits used the immunoreactive probe data from 20 trial participants associated with samples at each of the four timepoints in one of the three vaccinated cohorts. The normalised IgG-binding values for a given probe were then fitted to a model of the form:

$$i_{p,t} = X_d d + X_t t + Z_S + \varepsilon$$

Where $i_{p,t}$ is the IgG binding to a probe $p$ at time $t$ (measured as timepoints one to four); $X_t$ is the fixed effect of increasing timepoint on IgG binding; $X_d$ is the fixed effect of vaccine dose on IgG binding, with $d$ corresponding to the dose in micrograms; $Z_s$ is the random effect associated with each trial participant $s$, and $\varepsilon$ is an error parameter. The likelihood of $X_d$ being non-zero was calculated through fitting an identical model without the $X_d d$ term, and comparing the likelihood ratio of the two models with an ANOVA test. The same approach was used to calculate the likelihood of $X_t$ being non-zero. In each case, the 1584 likelihood values associated with all immunoreactive probes, or the 1384 non-DCL immunoreactive probes, were subject to a Benjamini-Hochberg correction using the R function p.adjust, and reported as significant if they were below the false discovery rate threshold of 0.05.

## Identification of protein features associated with immunogenicity

To identify those features of proteins that were associated with vaccine-induced IgG responses, a generalised linear model was fitted to the outcome, represented by a binary variable denoting whether a protein encoded a probe that was associated with a significant elevation in IgG binding by either the eBayes or LMM analyses (*Supplementary file 2*), and the explanatory variables, which described protein features. These included a continuous variable representing the mean length of the coding sequences in the corresponding cluster of orthologous genes (*Croucher et al., 2013a*), and binary variables corresponding to the presence of domains, identified by Pfam (*Punta et al., 2012*); a signal peptide, identified by SignalP (*Petersen et al., 2011*); transmembrane helices, identified by TMHMM (*Krogh et al., 2001*); and a lipoprotein processing motif, identified by Prosite (*Sigrist et al., 2013*). The proteins included in the analysis were the 1605 for which at least 90% sequence identity between the array and RM200 proteins were observed (*Supplementary file 1*); once the penicillin-binding protein and DCL variants were replaced with individual representatives of

the corresponding loci, this left a final dataset of 1600 proteins. Of these, 64 were deemed to be linked to an increase in IgG binding. The features included in the analysis were limited to those present in at least five proteins remaining in this dataset. The analysis was run in R (*R Core Team, 2017*) using the safeBinaryRegression package (*Konis, 2013*) until convergence; the stepAIC function of the MASS package (*Venables and Ripley, 2002*) was then used to refine the model, allowing the stepwise search to run in both directions, to produce the results shown in *Supplementary file 3*.

For the functional categorisation shown in *Figure 4C*, the annotation based on pre-vaccination data was used to describe ABTs (*Croucher et al., 2017*), and the set of proteins identified by either the LMM or eBayes analyses were used for WCV antigens. For both datasets, highly similar alleles split on functional information, such as the penicillin-binding proteins, were merged into single entries, whereas divergent alleles identified as separate COGs, such as the PclA variants, were kept as individual datapoints. The four DCL were each treated as a single set of orthologues that were immunogenic in both sets of proteins. Such classification resulted in datasets of 98 pre-vaccination ABTs and 74 WCV antigens, in a manner that was consistent between these groupings.

## Acknowledgements

We thank the volunteers of the Phase I trial of *S. pneumoniae* whole cell vaccine. Research reported in this publication was supported by The Bill and Melinda Gates Foundation and PATH. NJC is funded by a Sir Henry Dale Fellowship, jointly funded by the Wellcome Trust and Royal Society (Grant 104169/Z/14/Z). ML's work was supported by NIH Grant R01AI066304.

## Additional information

### Competing interests

Marc Lipsitch: Reviewing editor, *eLife*. Joseph J Campo, Timothy Q Le, Jozelyn V Pablo, Christopher Hung, Andy A Teng: are employees of Antigen Discovery, Inc. William P Hanage, Nicholas J Croucher: were supported by consulting payments from Antigen Discovery, Inc to work on this project. Xiaowu Liang: is an employee of Antigen Discovery, Inc and has an equity interest in Antigen Discovery, Inc. Richard Malley: has received honoraria or consulting fees from Merck and Affinivax, and has received research grants through his institution from PATH, the Bill and Melinda Gates Foundation, and Pfizer. The other authors declare that no competing interests exist.

### Funding

| Funder | Grant reference number | Author |
|---|---|---|
| Bill and Melinda Gates Foundation | | Joseph J Campo Timothy Q Le Jozelyn V Pablo Christopher Hung Andy A Teng |
| National Institutes of Health | R01AI066304 | Marc Lipsitch |
| Wellcome | 104169/Z/14/Z | Nicholas J Croucher |
| Royal Society | 104169/Z/14/Z | Nicholas J Croucher |
| PATH | | Andrea Tate Mark R Alderson |

The funders had no role in study design, data collection and interpretation, or the decision to submit the work for publication.

### Author contributions

Joseph J Campo, Conceptualization, Data curation, Software, Formal analysis, Investigation, Writing—original draft, Project administration, Writing—review and editing; Timothy Q Le, Jozelyn V Pablo, Christopher Hung, Andy A Teng, Hervé Tettelin, William P Hanage, Formal analysis, Writing—review and editing; Andrea Tate, Mark R Alderson, Conceptualization, Writing—review and

editing; Xiaowu Liang, Conceptualization, Formal analysis, Writing—original draft, Project administration, Writing—review and editing; Richard Malley, Conceptualization, Writing—original draft, Writing—review and editing; Marc Lipsitch, Conceptualization, Formal analysis, Writing—original draft, Writing—review and editing; Nicholas J Croucher, Data curation, Formal analysis, Investigation, Visualization, Writing—original draft, Writing—review and editing

### Author ORCIDs
Marc Lipsitch http://orcid.org/0000-0003-1504-9213
Nicholas J Croucher http://orcid.org/0000-0001-6303-8768

### Ethics
Clinical trial registration: The trial was registered with ClinicalTrials.gov with Identifier NCT01537185; the results are available from https://clinicaltrials.gov/ct2/show/results/NCT01537185.
Human subjects: The VAC-002 phase 1 study (ClinicalTrials.gov identifier: NCT01537185) was reviewed and approved by the Western Institutional Review Board and conducted in compliance with the study protocol, international standards of Good Clinical Practice and the Declaration of Helsinki. Participants were healthy adults aged 18 to 40 years at the time of consent, and had no evidence of chronic health issues, and nor any history of invasive pneumococcal disease or pneumococcal vaccination.

### Decision letter and Author response
Decision letter https://doi.org/10.7554/eLife.37015.041
Author response https://doi.org/10.7554/eLife.37015.042

## Additional files

### Supplementary files
• Supplementary file 1. Relationship between sequences on the proteome array and those in RM200. The 2190 proteins on the array selected based on the pneumococcal population in Massachusetts (*Croucher et al., 2013b*) are annotated and functionally classified in the first four columns, as described previously (*Croucher et al., 2017*). The four columns on the right show the distribution of these protein in *S. pneumoniae* RM200, the strain included in the WCV, as well as representatives of two closely-related species: *S. mitis* B6, and *S. pseudopneumoniae* IS7493. Where a protein is absent from a genome, there is a dash in the corresponding cell; otherwise, the sequence identity calculated from a pairwise alignment of protein sequences with MAFFT is shown.
DOI: https://doi.org/10.7554/eLife.37015.032

• Supplementary file 2. Significant changes in IgG binding identified by empirical Bayes and linear mixed effects model analyses. Each row corresponds to a probe on the proteome array associated with a significant change in IgG binding following WCV administration. The first six columns describe the functional annotation and classification of the corresponding protein, as well as whether it was categorised as an antibody-binding target (ABT) on the basis of high IgG binding in the pre-vaccination sample (*Croucher et al., 2017*). The next six columns contain statistics from the empirical Bayes analysis, in cases where there was a significant difference in $\Delta_{0\to84}$ between cohort three and the placebo group; otherwise, the cells contain 'NA'. These numbers describe the comparison of $\Delta_{0\to84}$ values between cohort three and the placebo group. The t statistic and B value, representing the log odds that the IgG binding differs between cohort three and the placebo group, are shown, along with the *p* values following individual tests, and after a Benjamini-Hochberg correction. The next four columns show the empirical Bayes statistics for an identical analysis in which all probes corresponding to the DCL were excluded. The next four columns show the output of the linear mixed effects model test for probes exhibiting a significantly increasing trend in IgG binding over the duration of the trial; otherwise, the cells contain 'NA'. The time coefficient describes the change in IgG binding over the trial in the vaccinated cohorts; only one probe has a negative coefficient, indicating vaccine doses reduced IgG binding over time. The table also shows calculation of a $\chi^2$ statistic as

part of a likelihood ratio test conducted against a linear mixed effects model with no time-dependent term, the resulting raw likelihood ratio, and the Benjamini-Hochberg corrected $p$ value. The final two columns show the likelihood and adjusted $p$ values for the same statistical test conducted with all DCL probes excluded; only the latter value differs between the two analyses.
DOI: https://doi.org/10.7554/eLife.37015.033

• Supplementary file 3. Protein features associated with elevated IgG responses following the administration of WCV. This multivariable logistic binary regression analysis fitted a model combining the explanatory variables of different protein characteristics to the binary dependent variable of whether or not a protein provoked an elevated IgG response, based on the probes listed in *Supplementary file 2*. The analysis removed variables preventing a maximum likelihood estimate, and the fitted model was refined by stepwise model selection based on Akaike information criterion (AIC) values. The table lists the features found to significantly associate with being identified as inducing a WCV-induced response: the protein's length, having a signal peptide for secretion, and possessing the listed functional motifs. The lipoprotein motif and SNP_bac_3 domains are associated with the solute-binding proteins of transporters, and the Transpeptidase domain is associated with cell wall metabolism proteins.
DOI: https://doi.org/10.7554/eLife.37015.034

• Transparent reporting form
DOI: https://doi.org/10.7554/eLife.37015.035

## Data availability

Sequencing data have been deposited in the ENA under accession code ERS2169631. Proteome array data analysed in this study is available as source data files for figures one and two.

The following dataset was generated:

| Author(s) | Year | Dataset title | Dataset URL | Database and Identifier |
|---|---|---|---|---|
| Tettelin H, Croucher N, Malley R | 2018 | *Streptococcus pneumoniae* RM200 Rx1E PdT *ΔlytA* | https://www.ebi.ac.uk/ena/data/view/ERS2169631 | European Nucleotide Archive, ERS2169631 |

The following previously published dataset was used:

| Author(s) | Year | Dataset title | Dataset URL | Database and Identifier |
|---|---|---|---|---|
| Croucher N | 2015 | Population genomic datasets describing the post-vaccine evolutionary epidemiology of *Streptococcus pneumoniae* | https://datadryad.org//resource/doi:10.5061/dryad.t55gq | Dryad Digital Repository, 10.5061/dryad.t55gq |

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
