## [Decision Letter]

Thank you for submitting your article "Panproteome-wide analysis of antibody responses to whole cell pneumococcal vaccination" for consideration by *eLife*. Your article has been reviewed by three peer reviewers, one of whom is a member of our Board of Reviewing Editors, and the evaluation has been overseen by Michel Nussenzweig as the Senior Editor. The following individual involved in review of your submission has agreed to reveal his identity: Jeremy Brown (Reviewer #3).

The reviewers have discussed the reviews with one another and the Reviewing Editor has drafted this decision to help you prepare a revised submission.

Summary:

In this submission, Croucher and colleagues describe a pan-proteome based analysis of antibodies generated in response to the whole cell pneumococcal vaccine (WCV) for *Streptococcus pneumoniae*. The authors first describe genetic differences between the WCV (derived from RM200) and the parental D39 strain using comparative genomics and describe various recombination and mutational events. Further analysis of SNPs between these strains provided useful information to describe the ontogeny of the current WCV. Next, the authors study antibody fingerprints in vaccinated individuals and report individual antibody fingerprints in their study participants but no distinct differences in the level of heterogeneity between cohorts of patients. Assessing specific antibody binding targets (ABTs), the authors demonstrate that vaccination with WCV resulted in an increase in IgG binding to targets that were enriched in individuals who had high levels of the targets prior to WCV vaccination. To further explore this, the authors interrogate the WCV strain for the presence of targets with orthologs in the progenitor RM200 strain. Next, the authors study antibody binding to specific antigens over different time points and at the beginning and end of the observation period. This analysis revealed 138 targets where IgG binging increased, these were derived from 112 protein sequences, which corresponded to 72 functionally distinct proteins. Thereafter, the authors describe responses to specific protein antigens and describe effects that seem to relate to the initial dose of the WCV. This is an interesting study, with potentially important implications.

Essential revisions:

All reviewers of the manuscript expressed concern that whilst the study is important and of high impact, it is presented a way that is not consistent with the broad readership base of *eLife*. There is extensive use of very technical terminology that will be difficult for the average reader to follow. Some text required repeated reading to understand the message. Figures could also be presented in a clearer and logical manner. Hence, the authors are requested to revise the manuscript to make it more accessible for *eLife* readers. Some specific comments are:

1) The manuscript needs a participant disposition flow chart, or a schedule of events that describes the patients and the time points. This should be the first figure in order for the reader to understand what samples are being analysed and which cohorts patients belonged to.

2) Clearly define the aims of the study in the Introduction. These are implicit in the Results and Discussion sections but it would help to have these explicitly defined in the Introduction.

3) Data presentation could be simplified to make it more readily accessible. Some specific points are:

A) The data (especially the box and whisker component) of Figure 5E and F are difficult to follow. Consider revising the presentation.

B) In Figure 5A to D; the compressed vertical access makes the figure harder to assess; this is partially true of many of the figures with multiple lines involved (Supplementary Figure 7 for example).

C) Supplementary file 2 is the crux of the data and the one that provides the specific data that the field will be interested in regards which antigens have increased antibody responses. Unfortunately, it is complex and full of statistical data, which makes it difficult to use. There is a strong case for inclusion of a simpler table (in the main text) listing the antigens to which there are increased antibody responses (and just a single line for the multiple allelic variants for the CBDs etc. as after all, there is only one version of each of these in the WCV), the strength of the increase in the antibody response, what proportion of subjects wherein this boost occurs, the D39 Spn gene number (for ease of interpreting what the corresponding gene is), what the pre-existing antibody level pre-vaccination was, and perhaps one or two highly relevant other parameters.

D) Some numbers that should be explicitly stated are: what proportion of ABTs have increased antibody responses to after vaccination with WCV; what proportion of non-ABTs have increased antibody responses to after vaccination with WCV; what proportion of SBPs have increased antibody responses to after vaccination with WCV.

E) There is a problem with the multiple allelic variants of ZmpA PspA etc. being included in the volcano plot. They actually represent boosting of responses by the single allelic variant in the WCV, hence having 8 or so dots representing the improved responses to multiple allelic variants is not a true representation of what is actually happening as an overview of the response to WCV. This perhaps might apply to some of the other figures leading to a skewing of some of the data, and this should be considered by the authors.

F) Figure 4—figure supplement Figure 14 did not seem to be very useful given most of these proteins are not in the WCV?

Other issues that should be addressed:

1) The assay for antibody binding measures an increase in fluorescence from a tagged antibody binding assay. As such, it does not exactly measure the level or functional capacity of the induced antibody responses. This does not need to be addressed by further experiments, as this is beyond the scope of the present study, but the authors need to discuss their interpretation of the increases in responses they observe and how this might translate into potential protective effects.

2) Some of the important implications are a little under-discussed. For example, the data shows which proteins stimulate a boost in antibody levels but this does not describe which of these responses are protective, which is a critical question. And why are some proteins boosting responses and other similar proteins do not? Is this to do with expression/quantity in the WCV? If there were data available on the latter that would be helpful to include. Identifying protective responses is not easy – correlations between individual antibody responses and functional assays of phagocytosis would be helpful but these data are probably not available and really would make a separate paper. Some mention in the Discussion would be useful.

3) The majority of the vaccine-induced responses were to solute binding and cell wall metabolism proteins. Antibody accessibility to such antigens may be limited by capsule, reducing any effect of antibody on systemic pneumococcal infection as was seen for anti-PsaA or PpmA in the study of Gor et al., 2005. However, numerous other studies have found that vaccination using PsaA is protective against nasopharyngeal carriage – this might reflect binding to the relatively less encapsulated transparent phase variants that bind more effectively to the nasopharyngeal epithelium. The authors should discuss the accessibility or otherwise of potentially protective antibodies in relation to capsule phase variation and nasopharyngeal carriage.

[Editors' note: further revisions were requested prior to acceptance, as described below.]

Thank you for resubmitting your work entitled "Panproteome-wide analysis of antibody responses to whole cell pneumococcal vaccination" for further consideration at *eLife*. Your revised article has been favorably evaluated by Michel Nussenzweig (Senior Editor), a Reviewing Editor, and two reviewers.

The manuscript has been improved but there are some remaining issues that need to be addressed before acceptance. We consulted another BRE at *eLife*, with statistical expertise, and following concerns emerged:

1) The previous decision letter communicated that the manuscript was not written in an accessible manner and that it needed to be simplified in order for readers to be able to follow what has been done. While the authors do seem to have addressed the specific recommendations, this had been done by adding to the text rather than reorganising the presentation of the manuscript. Please give this further attention, readability to a broad audience is important for *eLife*.

2) Related to the point above, some aspects are difficult to follow. The Abstract suggests this is a phase 1 vaccine clinical trial but it is really a descriptive analysis of antibody responses in the longitudinal measurements of the trial participants. (This is acknowledged in the subsection “Study design and participant”.) The Abstract states that 72 functionally distinct proteins were associated with increases but this result is difficult to isolate in the main manuscript. The Abstract needs to communicate that this is a descriptive analysis.

3) The trial was designed for Phase 1 safety outcomes so this study is making use of existing samples and will be under-powered for many questions that the researchers may wish to address. Consider this point and address it in your manuscript.

4) Very few statistical analyses are presented though hypothesis tests and statistical significance has clearly been used to interpret results. Further elaboration on tests would assist in understanding the data.

5) Further statistical analysis concerns:

A) Subsection “Stable antibody “fingerprints” of vaccinated individuals”, last paragraph. A Kruskall Wallis (non-parametric) test has been used on a sample size of 67,947 for the per probe rather than per individual analysis. With a sample size so large a parametric analysis could have been used. Large sample sizes such as these will be able to detect very small differences that may not be biologically important. A parametric analysis would yield mean values per group to aid understanding. This analysis seems to be entirely inappropriate as all correlations have been ignored. A mixed effects linear model would have correctly taken account of the correlations.

B) Subsection “WCV elicits elevated IgG to a specific minority of proteins”, second paragraph. It also seems odd to use Fisher's exact test when the sample size is so large. It also seems inappropriate to assume independence here.

C) Subsection “Identification of specific antigens”, second paragraph. A BH correction for multiple testing has been adopted but it is not clear how this was done. How many tests were performed and what level was the FDR controlling for? Wherever corrected p-values are presented the full procedure needs to be documented.

6) Figure 7 needs to show the number on each graph, and identify what the colours represent.

---

## [Author Response]

Essential revisions:All reviewers of the manuscript expressed concern that whilst the study is important and of high impact, it is presented a way that is not consistent with the broad readership base of eLife. There is extensive use of very technical terminology that will be difficult for the average reader to follow. Some text required repeated reading to understand the message. Figures could also be presented in a clearer and logical manner. Hence, the authors are requested to revise the manuscript to make it more accessible for eLife readers.

On revisiting the manuscript, we agree with the reviewers that sections of the Results were not clearly written, and therefore we have comprehensively redrafted those parts which we felt could be improved. We have also ensured terms such as probes, solute binding proteins and diverse core loci are now explicitly defined, and have provided a brief description of the t-SNE, empirical Bayes and linear mixed effects model methodologies at the relevant points in the manuscript.

1) The manuscript needs a participant disposition flow chart, or a schedule of events that describes the patients and the time points. This should be the first figure in order for the reader to understand what samples are being analysed and which cohorts patients belonged to.

This is a very helpful suggestion. We have now added such a flowchart as a new Figure 1. This details the design of the trial, the scheduling of the vaccine doses and serum collection, and the samples available for analysis.

2) Clearly define the aims of the study in the Introduction. These are implicit in the Results and Discussion sections but it would help to have these explicitly defined in the Introduction.

The reviewers are correct that the Introduction ended somewhat abruptly, without defining the aims of the study, in the previous version of the manuscript. Therefore we have now appended a section outlining the planned analyses to the end of the Introduction: “This study therefore aimed to identify the range and types of antigens to which IgG responses were mounted following WCV administration, and how these might vary between individuals. […] These analyses should provide information on whether the WCV is likely to consistently induce antibodies capable of recognising antigens conserved across *S. pneumoniae* isolates.”.

3) Data presentation could be simplified to make it more readily accessible. Some specific points are:A) The data (especially the box and whisker component) of Figure 5E and F are difficult to follow. Consider revising the presentation.

We accept this presentation was not ideal in the submitted version of the manuscript. Hence we have now moved these plots from Figure 5, to allow for the expansion of panels A-D, and created a new Figure 6. Panels A and B of Figure 6 reproduce the previous scatterplots, but with the box and whisker plots removed. Instead, the distributions of initial IgG binding values for those probes showing an increase between the relevant timepoints are represented as histograms in the new panels C and D.

B) In Figure 5A to D; the compressed vertical access makes the figure harder to assess; this is partially true of many of the figures with multiple lines involved (Supplementary Figure 7 for example).

This was certainly a deficiency of the original graphs, and we have now adjusted the vertical axis range of Figure 5 panels A-D to display the data more clearly, as well as removing two panels from Figure 5 to create a separate Figure 6, to allow panels A-D to be expanded. We have also split Supplementary Figure 7 into four separate figures (Figure 4—figure supplements 8, 9, 12 and 15), to make these plots easier for the reader to view.

C) Supplementary file 2 is the crux of the data and the one that provides the specific data that the field will be interested in regards which antigens have increased antibody responses. Unfortunately, it is complex and full of statistical data, which makes it difficult to use. There is a strong case for inclusion of a simpler table (in the main text) listing the antigens to which there are increased antibody responses (and just a single line for the multiple allelic variants for the CBDs etc. as after all, there is only one version of each of these in the WCV), the strength of the increase in the antibody response, what proportion of subjects wherein this boost occurs, the D39 Spn gene number (for ease of interpreting what the corresponding gene is), what the pre-existing antibody level pre-vaccination was, and perhaps one or two highly relevant other parameters.

This is a good suggestion, and accordingly we have now generated Table 1 to summarise the increases in IgG binding associated with the WCV antigens. This links each of the proteins to which a significant increase in IgG binding was detected to the relevant *S. pneumoniae* D39 coding sequence, where possible, and aggregates data across variants for each of the DCL. The table also provides the median IgG binding at the start of the trial, as well the median change by the end of the trial in each cohort; all of these values are associated with interquartile ranges, to assess the variation in the data within cohorts. These ranges are in place of a proportion of individuals in whom an increase above a specific threshold was observed, as the numbers within each cohort are relatively small, and therefore variation in the proportion exhibiting a rise above a given value could be difficult to interpret. This format is intended to provide all the information requested by the reviewers in as accessible a form as is possible.

D) Some numbers that should be explicitly stated are: what proportion of ABTs have increased antibody responses to after vaccination with WCV; what proportion of non-ABTs have increased antibody responses to after vaccination with WCV; what proportion of SBPs have increased antibody responses to after vaccination with WCV.

The reviewers are absolutely right that these numbers should have been more clearly stated in the text. We have now added the results of the suggested analysis. Where a cutoff was used to identify probes associated with significantly elevated IgG binding (Figure 2), we now provide the per-probe statistics: the 325 probes from ABTs accounted for 23 of 47 (48.9%), 17 of 43 (39.5%), and 60 of 129 (46.5%) probes associated with significant increases in cohorts one to three, respectively. All of these were significant by Fisher’s exact tests.

For the 112 proteins found to be significantly associated with IgG rises by the empirical Bayes and linear mixed effects model tests, 71 were ABT proteins (63.4%); hence 137 of the 208 did not show such an increase. Of the 1,235 immunoreactive non-ABTs, 41 were identified as eliciting a significant increase in IgG binding (3.32%), and 1,194 were not. Therefore, there is a significant enrichment of ABTs in the antigens eliciting a response to the WCV (Fisher’s exact test, odds ratio = 15.0, 95% confidence interval 9.68-23.6, *p* < 2.2x10^-16^). When considering only the 1,062 immunoreactive proteins represented on the array and exhibiting at least 90% similarity with a protein expressed by the WCV, ABTs accounted for 39 of the 71 proteins associated with a significant increase in IgG binding (54.9%). Thirty-four of the ABTs conserved in the WCV were not associated with a significant increase, compared with 957 non-ABTs. Hence ABTs represented only 3.43% of the conserved proteins not exhibiting a significant post-WCV rise in IgG binding. This again corresponds to a significant enrichment (Fisher’s exact test, odds ratio = 33.9, 95% confidence interval = 18.4-63.6, *p* < 2.2x10^-16^).

Regardless of IgG binding, using the Pfam domains now described in the Methods, we identified 26 SBPs in the *S. pneumoniae* RM200 genome; manual annotation added AliA to bring the total to 27. Of these, 25 were immunoreactive and represented on the proteome array; correspondingly, there are 1,922 non-SBPs encoded by the RM200 genome, of which 1,037 were represented on the array and immunoreactive. Eleven SBPs (44.0% of the immunoreactive SBPs on the array), and 60 non-SBPs (5.79% of the non-SBPs that were immunoreactive and represented on the array), were associated with significant increases in IgG binding post-WCV. Hence SBPs were strongly enriched in those proteins triggering an immune response (Fisher’s exact test, odds ratio = 12.7, 95% confidence interval = 5.00 – 31.6, *p* = 9.9x10^-8^).

These SBPs therefore accounted for 11 of the 71 ABTs associated with a significant increase in IgG binding (15.5%). Of the other 137 ABTs not showing such an increase, only six were SBPs (4.38%). Hence there was a significant enrichment for SBPs within the ABTs provoking a further IgG response (Fisher’s exact test, odds ratio = 3.97, 95% confidence interval 1.28-13.7, *p =* 0.014).

For completeness, we have also added the same statistics for the cell wall synthesis and processing machinery. As this set of proteins do not share a common structural motif that allows them to be consistently identified from the genome, the enrichment of these proteins relative to other ABTs was tested. Overall, 16 of the 25 cell wall machinery ABTs were among the 71 proteins triggering a significant immune response by eBayes or LMM (64.0%). This corresponded to a 3.43-fold enrichment in the ABTs associated with elevated IgG binding (Fisher’s exact test, odds ratio = 4.11, 95% confidence interval = 1.60-11.2, *p* = 0.0014).

We have now provided all of these statistics in the main Results section, and the transparent reporting form. We have chosen to omit the odds ratios, and associated confidence intervals, because these imply uncertainty associated with sampling, rather than a comparison of probe populations for which we have almost complete data. Instead, we report the proportions, fold enrichments and *p* values.

E) There is a problem with the multiple allelic variants of ZmpA PspA etc. being included in the volcano plot. They actually represent boosting of responses by the single allelic variant in the WCV, hence having 8 or so dots representing the improved responses to multiple allelic variants is not a true representation of what is actually happening as an overview of the response to WCV. This perhaps might apply to some of the other figures leading to a skewing of some of the data, and this should be considered by the authors.

The reviewers are right that this is a difficult aspect of the data to address. In terms of the volcano plot, these are the responses to proteins on the array that are detected and used in normalisation procedures, and therefore from a technical perspective it would be misleading to omit them. Also, from a biological perspective, we feel it is interesting that the response to a single variant of these diverse proteins triggers antibodies capable of recognising multiple other variants, even though these correspond to distinct representatives of the overall diversity at these loci. Hence a volcano plot including these points is the more accurate representation of the immune response to the WCV.

However, we appreciate the reviewers’ concerns about how this over-representation of DCL sequences may affect the statistical analyses. These probes were excluded from some analyses in the manuscript where it was clearly appropriate, such as that comparing ABTs present in the WCV with those absent from it, and we have made this clearer in the text. To ascertain their impact on the identification of immunogenic proteins within the WCV, we have repeated the empirical Bayes and linear mixed effects model analyses with the probes corresponding to PspA, PspC, ZmpA and ZmpB excluded. This resulted in four of the 78 non-DCL loci probes no longer being significant in either test, corresponding to proteins Pbp1B, RexA, YneF, and a PclA allele. Given the small number of changes, and the distribution of the discrepancies between multiple functional categories, we feel this demonstrates the conclusions of the paper are generally robust to the inclusion of the DCL. However, it is important to note the dependency of the output on the inclusion of the DCL alleles in the text, and we have also added the outputs of the tests excluding the DCL to Supplementary file 2. The accompanying volcano plot, omitting the DCL, is now also included as a supplementary figure (Figure 4—figure supplement 6).

F) Figure 4—figure supplement 14 did not seem to be very useful given most of these proteins are not in the WCV?

We think Figure 4—figure supplement 14 is helpful in demonstrating the specificity of the immune response and proteome array, particularly with regard to the different alleles of PclA. Furthermore, not only do these immunogenic proteins absent from the WCV effectively serve as a negative control, confirming the responses to ABTs are generally specific to those in the WCV, they also represent candidate vaccine antigens likely to be included in multiprotein or conjugate formulations. Therefore it is likely that responses to these antigens will be of interest in future studies, and so it may be helpful to explicitly plot these data for comparison with cases where individuals are exposed to these proteins.

Other issues that should be addressed:1) The assay for antibody binding measures an increase in fluorescence from a tagged antibody binding assay. As such, it does not exactly measure the level or functional capacity of the induced antibody responses. This does not need to be addressed by further experiments, as this is beyond the scope of the present study, but the authors need to discuss their interpretation of the increases in responses they observe and how this might translate into potential protective effects.

This is certainly an important point to convey to allow the reader to fully interpret the results. We have therefore added a section to the Discussion, explicitly making the point raised by the reviewers: “The surface-associated proteins enriched among WCV antigens may be important components of a protective IgG response, as recognition of these structures enables intact S*. pneumoniae* cells to be agglutinated (Mitsi et al., 2017) or targeted for opsonophagocytosis (Hyams et al., 2010). […] It is not clear whether a threshold correlate of protection, as determined for anti-capsular antibodies induced by PCVs (Andrews et al., 2014), could be applicable to a broad range of distinct protein antigens.”

2) Some of the important implications are a little under-discussed. For example, the data shows which proteins stimulate a boost in antibody levels but this does not describe which of these responses are protective, which is a critical question. And why are some proteins boosting responses and other similar proteins do not? Is this to do with expression/quantity in the WCV? If there were data available on the latter that would be helpful to include. Identifying protective responses is not easy – correlations between individual antibody responses and functional assays of phagocytosis would be helpful but these data are probably not available and really would make a separate paper. Some mention in the Discussion would be useful.

The reviewers raise multiple interesting points. Ultimately, we do not know exactly why some proteins trigger IgG responses, but others do not, despite being similar in their structural and functional characteristics, or pre-vaccine levels of IgG binding. There appears to be some pattern regarding those proteins that are typically subcapsular eliciting increases in antibody binding, whereas the adhesins and degradative proteins, which usually extend beyond the capsule, do not contribute much to the overall response. The suggestion that this could relate to differences in the levels at which proteins are present in the WCV, relative to live colonising *S. pneumoniae*, is an excellent one, to which we now refer in the relevant section we have added to the Discussion: “The reasons for this heterogeneity are unclear. […] It is more conceivable that many ABTs may not have been expressed by the RM200 cells during culture, although the strong and apparently novel response to some predominantly intracellular proteins, such as FlrR, suggests expression of surface-associated proteins would have to be very low for lack of exposure to explain the absence of an immune response.”.

We also retain our previous discussion of alternative possible explanations, including saturation of the measurements from the array. We do not have information on whether these antibodies responses are functionally protective, and we hope the text added in response to the previous point is sufficient to make this clear to the reader.

3) The majority of the vaccine-induced responses were to solute binding and cell wall metabolism proteins. Antibody accessibility to such antigens may be limited by capsule, reducing any effect of antibody on systemic pneumococcal infection as was seen for anti-PsaA or PpmA in the study of Gor et al., 2005. However, numerous other studies have found that vaccination using PsaA is protective against nasopharyngeal carriage – this might reflect binding to the relatively less encapsulated transparent phase variants that bind more effectively to the nasopharyngeal epithelium. The authors should discuss the accessibility or otherwise of potentially protective antibodies in relation to capsule phase variation and nasopharyngeal carriage.

This is an excellent point, and therefore we have added a short section addressing this point in the Discussion: “Therefore, these antibodies could afford broad protection against a diverse set of strains, as long as these antibodies can penetrate the capsular envelope to bind their cognate proteins. […] These are more commonly associated with colonising isolates adhering to the nasopharyngeal epithelium, rather than those causing disease.”

[Editors' note: further revisions were requested prior to acceptance, as described below.]

The manuscript has been improved but there are some remaining issues that need to be addressed before acceptance. We consulted another BRE at eLife, with statistical expertise, and following concerns emerged:1) The previous decision letter communicated that the manuscript was not written in an accessible manner and that it needed to be simplified in order for readers to be able to follow what has been done. While the authors do seem to have addressed the specific recommendations, this had been done by adding to the text rather than reorganising the presentation of the manuscript. Please give this further attention, readability to a broad audience is important for eLife.

We appreciate the importance of making the manuscript accessible to a broad audience, and have sought to improve on the edits already made in response to the suggestions of the original set of reviewers. Having re-read the text, we have now reorganised the Results section into a larger number of smaller sections, with headers added to guide the reader through the main conclusions. We have also comprehensively reworded some parts we identified as being dense with technical language, including clarifying the selection and design of some of the statistical tests, as detailed below. We hope this suitably improves the accessibility of the material.

2) Related to the point above, some aspects are difficult to follow. The Abstract suggests this is a phase 1 vaccine clinical trial but it is really a descriptive analysis of antibody responses in the longitudinal measurements of the trial participants. (This is acknowledged in the subsection “Study design and participant”.) The Abstract states that 72 functionally distinct proteins were associated with increases but this result is difficult to isolate in the main manuscript. The Abstract needs to communicate that this is a descriptive analysis.

We agree it is helpful to make it clear that these data are not the primary output of the trial, although still feel it is important to specify the samples originate from placebo-controlled trial, rather than an alternative type of clinical study (e.g. case control or observational study). As all phase 1 trial outcomes are inherently descriptive, this distinction is not necessarily clear-cut. We have therefore changed the relevant sentence in the Abstract from:

“Immunoglobulin G (IgG) responses to a WCV were quantified by applying longitudinally-sampled sera from 35 adults in a placebo-controlled phase I trial to a panproteome microarray…” to: “Immunoglobulin G (IgG) responses to a WCV were characterised by applying longitudinally-sampled sera, available from 35 adult placebo-controlled phase I trial participants, to a panproteome microarray”.

We have also made the sub-sampling from the trial clear in the Results section by editing the text to read, “Overall, 130 samples were studied from 35 of the 42 trial participants”.

The identification of 72 functionally-distinct proteins requires the combination of multiple analyses (assessing the most useful statistical tests; identifying immunogenicity using these tests, then functionally annotating the immunogenic probes). Therefore this result is only presented once these preceding analyses have been introduced in a stepwise manner, to allow readers to understand how we arrived at this conclusion.

3) The trial was designed for Phase 1 safety outcomes so this study is making use of existing samples and will be under-powered for many questions that the researchers may wish to address. Consider this point and address it in your manuscript.

It is true there are inherent limitations to samples from a small trial, which we now acknowledge in the Discussion (“The phase I clinical trial of WCV was primarily designed to measure safety outcomes, thus statistical power to assess immunogenicity at the proteome scale may have been limited.”). There is no particular immunological endpoint for which we are necessarily underpowered due to only using a subset of trial participants, as no specific power analysis was conducted prior to the trial (see Materials and methods), and the panproteome array technology was not included in the original design of the trial. In general, as we were able to identify many immunogenic proteins, it appears we were appropriately powered to detect the most biologically-significant immune responses. Those proteins found to be non-immunogenic appear to genuinely fail to trigger a detectable response (e.g. Figure 4—figure supplements 8, 9, 12 and 15), rather than a smaller response we were underpowered to identify as significant, suggesting the sample size was not necessarily restrictive.

We do not think it would be helpful to speculate on additional hypotheses we could test with larger trials. Indeed, the greater limitation on data interpretation is the use of adults, rather than infants, as already mentioned in the Discussion.

4) Very few statistical analyses are presented though hypothesis tests and statistical significance has clearly been used to interpret results. Further elaboration on tests would assist in understanding the data.

After reviewing the presentation of the statistical tests in the manuscript, we agree with the reviewer that some tests could be more thoroughly explained. Therefore we have added text to improve the description of the hypotheses being tested, and statistical approach employed, in three sections:

- We have clarified the hypothesis being tested with Fisher’s exact tests of association between antibody-binding targets and vaccine antigens.

- We have explicitly stated the motivation for the multivariable analysis of protein characteristics.

- We have more thoroughly explained the comparison of pre-vaccination IgG binding to antigens associated with early, or late, vaccine-induced antibody responses.

5) Further statistical analysis concerns:A) Subsection “Stable antibody “fingerprints” of vaccinated individuals”, last paragraph. A Kruskall Wallis (non-parametric) test has been used on a sample size of 67,947 for the per probe rather than per individual analysis. With a sample size so large a parametric analysis could have been used. Large sample sizes such as these will be able to detect very small differences that may not be biologically important. A parametric analysis would yield mean values per group to aid understanding. This analysis seems to be entirely inappropriate as all correlations have been ignored. A mixed effects linear model would have correctly taken account of the correlations.

The reviewer is correct that a linear mixed effects model better accounts for within-individual correlations, and therefore we have replaced this Kruskal-Wallis test with such an analysis. Linear mixed effects models were fitted with trial participant as a random effect, and either cohort and probe, or just probe, as fixed effects. An ANOVA analysis comparing these two fits found the inclusion of cohort significantly improved the model fit (χ^2^ = 8.96, df = 3, p = 0.030). When this model fitting was repeated using just the immunoreactive probes (exhibiting a maximal IgG binding at least double that of the background level), excluding probes associated with IgG binding levels unlikely to be biologically significant, this comparison found a more statistically significant improvement when cohort was included (χ^2^ = 11.8, df = 3, p = 0.00807). This indicates there is significant variation between cohorts at the level of individual probes, consistent with the previous Kruskal-Wallis test.

B) Subsection “WCV elicits elevated IgG to a specific minority of proteins”, second paragraph. It also seems odd to use Fisher's exact test when the sample size is so large. It also seems inappropriate to assume independence here.

Fisher’s exact test for association between variables using a contingency table is applicable to any sample size in principle, but is sometimes limited in practice due to computational feasibility. These values were easily calculated, and therefore there was no need to employ a less accurate test, such as the Chi squared test; analysis with this alternative test gave very similar p values, which were typically marginally less conservative than those of the Fisher’s exact tests. Therefore we have retained the precise and conservative outputs of the Fisher’s exact test in the manuscript.

The reviewer is correct that the two variables being tested (whether or not the probe represented an antibody-binding target, and whether or not the probe increased IgG binding by more than a threshold value) are not completely independent. However, the lack of independence is conservative with respect to the conclusion of the test. Antibody-binding targets have high initial antibody binding levels (Croucher et al., 2017), and in the absence of vaccination, regression to the mean results in a negative correlation between pre-vaccination IgG binding and the post-vaccination change in IgG binding (Figure 3A). Additionally, detection of post-vaccination increases in IgG binding would more likely be limited by saturation for probes starting at high IgG binding pre-vaccination; however, the observed levels of IgG binding in this experiment means there was little, if any, effect of saturation. As both of these effects result in a tendency to associate non-ABTs with greater increases in IgG binding than ABTs, whereas we concluded the opposite, we consider this result robust, and the methodology appropriate. This concurs with the consistency of the threshold-based analysis with the eBayes and LMM analyses later in the manuscript (Figure 4—figure supplement 7).

C) Subsection “Identification of specific antigens”, second paragraph. A BH correction for multiple testing has been adopted but it is not clear how this was done. How many tests were performed and what level was the FDR controlling for? Wherever corrected p-values are presented the full procedure needs to be documented.

We have now specified the p values for the 1,584 immunoreactive probes (reduced to 1,384 probes when the diverse core loci were excluded) were subject to a standard Benjamini-Hochberg correction, using either the base R function p.adjust, or the topTable function from limma. Results were considered significant if they were below the false discovery rate threshold of 0.05. All this information has now been added to the Results and Materials and methods sections.

6) Figure 7 needs to show the number on each graph, and identify what the colours represent.

Thank you for identifying these deficiencies in the legend of Figure 7. We have now annotated the total number of scatterpoints (n = 414), explaining the in the legend that this corresponds to one per immunogenic probe (n = 138) in each of the three vaccinated cohorts. The colours associate datapoints with cohorts, consistent with the previous figures, as now explained in the legend and an added key. The numbers of datapoints represented by the histograms in panels C and D are also now marked on the revised figure, and included in the legend.